# Lightweight Protocols for Distributed Private Quantile Estimation

**Anders Aamand** [1]   **Fabrizio Boninsegna** [2]   **Abigail Gentle** [3]   **Jacob Imola** [1]   **Rasmus Pagh** [1]

## Abstract

Distributed data analysis is a large and growing field driven by a massive proliferation of user devices, and by privacy concerns surrounding the centralised storage of data. We consider two *adaptive* algorithms for estimating one quantile (e.g. the median) when each user holds a single data point lying in a domain $[B]$ that can be queried once through a private mechanism; one under local differential privacy (LDP) and another for shuffle differential privacy (shuffle-DP). In the adaptive setting we present an $\varepsilon$-LDP algorithm which can estimate any quantile within error $\alpha$ only requiring $O(\frac{\log B}{\varepsilon^2 \alpha^2})$ users, and an $(\varepsilon, \delta)$-shuffle DP algorithm requiring only $\widetilde{O}((\frac{1}{\varepsilon^2} + \frac{1}{\alpha^2}) \log B)$ users. Prior (nonadaptive) algorithms require more users by several logarithmic factors in $B$. We further provide a matching lower bound for adaptive protocols, showing that our LDP algorithm is optimal in the low-$\varepsilon$ regime. Additionally, we establish lower bounds against non-adaptive protocols which paired with our understanding of the adaptive case, proves a fundamental separation between these models.

## 1. Introduction

A strong trend in recent years has been towards *federated* computations (Kairouz et al., 2021) in which algorithms are run on a distributed dataset rather than by collecting data and performing the computation in a centralized manner. This trend is motivated by the wish to protect individuals' data, as well as organizations' wish steer clear of liability issues stemming from collecting and handling private data. The

leading approach to federated computations with *formal* privacy guarantees is to use differential privacy (DP) (Dwork et al., 2006) which limits the amount of information that can be inferred about a given user's input by selecting the output from a suitable probability distribution defined by the inputs. A particularly simple and appealing setup is *local differential privacy* (LDP), in which each user individually sends the output of a differentially private algorithm to a central "analyzer", who in turn uses all the user outputs to approximate a function of the inputs. Though LDP is not the only approach to federated computations with differential privacy, it has been influential. For example, LDP has been used in industrial deployments of differential privacy (Cormode et al., 2018; Erlingsson et al., 2014; Differential Privacy Team, 2017), and there is a rich theory showing both upper and lower bounds on the privacy-utility trade-offs that are possible under LDP. When giving privacy guarantees under LDP it is common to consider *pure* differential privacy since it is known that any non-adaptive protocol satisfying approximate differential privacy can be converted into an equivalent one satisfying pure differential privacy (Bun et al., 2019). An interesting aspect of LDP algorithms is that they can be used as building blocks of more sophisticated algorithms offering better privacy utility trade-offs (with stronger trust assumptions), for example in the *shuffle model* (Bittau et al., 2017; Cheu et al., 2018; Erlingsson et al., 2019).

**Quantile Estimation**   In this paper we study the problem of *quantile estimation* under local differential privacy constraints. Specifically, we consider the setting of a dataset $X$ of $n$ datapoints $x_1, \ldots, x_n \in [B] := \{1, \ldots, B\}$, where $B$ is an integer parameter. We note that some works on quantile estimation consider another setting where the input points are real numbers, e.g., from $[0, 1]$, but this requires assumptions on the data distribution – in Appendix H, we will discuss how our algorithms can be brought to work in the continuous setting under mild assumptions on the distribution of the input data. Given the dataset $X$, we define the empirical CDF $F_X : [B] \to [0, 1]$ by

$$F_X(i) = \frac{1}{n}|\{j \in [n] \mid x_j \leq i\}|, \tag{1}$$

that is, $F_X(i)$ is the fraction of elements in the dataset that are smaller than or equal to $i$. Given $q \in (0, 1)$ we would

---

[1]BARC and Department of Computer Science, University of Copenhagen, Denmark [2]Department of Information Engineering, University of Padova, Italy [3]School of Computer Science, University of Sydney, Australia. Correspondence to: Anders Aamand <aa@di.ku.dk>, Abigail Gentle <abigail.gentle@sydney.edu.au>, Jacob Imola <jaim@di.ku.dk>, Fabrizio Boninsegna <fabrizio.boninsegna@phd.unipd.it>.

*Proceedings of the 42ⁿᵈ International Conference on Machine Learning*, Vancouver, Canada. PMLR 267, 2025. Copyright 2025 by the author(s).

ideally like to output an approximate $q$th quantile of the dataset, that is, a value $m \in [B]$ such that $F_X(m)$ is approximately equal to $q$. Such a value $m$ may not always exist, for example if all $x_i$ are equal. Instead, we measure the approximation guarantee in terms of a parameter $\alpha \in (0, 1)$ and we are happy to report a value $m$ such that $q$ is contained in the interval $[F_X(m) - \alpha, F_X(m+1) + \alpha]$.

**Adaptive Local Differential Privacy**   In this paper we consider LDP algorithms that work in rounds, making *adaptive* choices of what information should be released in each round. An adaptive LDP protocol involves $n$ users indexed by $i = 1, \ldots, n$, with user $i$ holding a data item $x_i$, and an *aggregator* that coordinates the protocol. In round $t$ the aggregator *queries* a set $I_t \subseteq \{1, \ldots, n\}$ of one or more users, asking them to run a differentially private mechanism $\mathcal{M}_t$ on their data. The output of $\mathcal{M}_t(x_i)$ is then sent to the aggregator for each $i \in I_t$. Protocols in this model can be adaptive in the sense that the choice of mechanism $\mathcal{M}_t$ can depend on the results of mechanisms in rounds $1, \ldots, t-1$. We consider *sequentially adaptive* protocols in which the query sets $I_1, I_2, \ldots$ are disjoint, such that the privacy guarantee for each user is simply determined by the privacy guarantee of the mechanism that was used for the LDP report on their data (if any). In contrast *non-adaptive* LDP protocols can be run in a single round. The private mechanism $\mathcal{M}_t$ is predetermined and does not depend on the outputs of $\mathcal{M}_1, \ldots, \mathcal{M}_{t-1}$. It is often the case that all $\mathcal{M}_t$ are the same. Adaptive mechanisms often offer much improved utility/privacy tradeoffs compared to their non-adaptive counterparts (Acharya et al., 2022; Joseph et al., 2020) but they are harder to coordinate and thus less desirable from a practical perspective.

For quantile estimation, each user $i$ holds the datapoint $x_i \in [B]$ and our goal is to estimate a quantile, with error described as above, such that $\mathcal{M}_t$ satisfies LDP. It is not hard to see that an algorithm for estimating the median, that is $q = 1/2$, can be used to estimate any quantile of the dataset with only a constant factor increase of the approximation guarantee. This is because we can reduce the general case to the median by introducing $n$ additional, virtual users holding data, $(1-q)n$ users each holding the value 1 and $qn$ users holding the value $B$ (see Lemma B.1). Thus, for our algorithm we focus on estimating the median. We refer to this problem with desired accuracy $\alpha$ and LDP privacy parameter $\varepsilon$ as LDPemp-median$(\{x_i\}_{i=1}^n, \alpha, \varepsilon)$ (see Section 2 for the formal definition). We derive a sequentially adaptive algorithm with the following guarantee:

**Theorem 1.1.** *For all $\alpha \in (0, \frac{1}{4})$, and $\varepsilon \in (0, 1)$, there exists a sequentially adaptive $\varepsilon$-LDP protocol solving* LDPemp-median$(\{x_i\}_{i=1}^n, \alpha, \varepsilon)$ *with probability at least $1 - \frac{1}{B}$ for any dataset with $n \geq c\frac{\log B}{\varepsilon^2 \alpha^2}$ for a universal constant $c$.*

The algorithm queries one user at a time (so each $|I_t| = 1$) and proceeds for $n$ rounds. In terms of communication and run time, our algorithm is efficient: each user communicates just 1 bit to the server, and each round has update time $O(\log B)$. In addition, we show that the error of our protocol is *optimal* up to constant factors under (sequentially-adaptive) LDP:

**Theorem 1.2.** *Suppose that $B$ is sufficiently large, $\alpha \leq \frac{1}{2}$, and $\varepsilon < 1$. Any sequentially adaptive LDP protocol solving* LDPemp-median$(\{x_i\}_{i=1}^n, \alpha, \varepsilon)$ *with probability at least $3/4$ for any dataset of size $n \geq n_0$ must have $n_0 = \Omega\left(\frac{\log B}{\varepsilon^2 \alpha^2}\right)$.*

*Remark:* The above theorem is stated for the median, but as we will see, the same lower bound holds for estimating any quantile $q \in (2\alpha, 1 - 2\alpha)$. Note that for $q \leq \alpha$ or $q \geq 1 - \alpha$, there is a trivial protocol that outputs either 1 or $B$. Combined with the above observation for reducing a general quantile to the median, our results therefore show that $\Theta(\frac{\log B}{\varepsilon^2 \alpha^2})$ is essentially the correct bound for quantile estimation under sequentially-adaptive LDP in the high privacy regime.

**Non-Adaptive Protocols**   Theorems 1.1 and 1.2 settle the optimal privacy/utility tradeoffs for adaptive LDP protocols in the high privacy regime. As we will discuss in Section 1.1, all non-adaptive mechanisms that we are aware of require a polylog$(B)$ factor more users to solve LDPemp-median, which can be significant as $B$ is typically a large parameter such as $2^{32}$. Our privacy/utility tradeoff in the adaptive case is therefore much better than for known non-adaptive protocols, but as discussed, non-adaptive protocols are more practically appealing. A natural question is thus if the gap is inherent. We settle this question in the positive essentially showing that *any* non-adaptive protocol must incur an additional logarithmic factor in $B$ in the number of users required for a desired accuracy. Thus, non-adaptivity, while practically desirable, comes at a significant price in utility. Our result is as follows.

**Theorem 1.3.** *Suppose $B$ is sufficiently large, $B^{-\Omega(1)} \leq \alpha \leq c$ for a universal constant $c$, and $\varepsilon \leq \frac{1}{\log(1/\alpha)}$. Suppose that there exists a non-adaptive $\varepsilon$-LDP algorithm solving* LDPemp-median$(\{x_i\}_{i=1}^n, \alpha, \varepsilon)$ *with probability at least $\frac{3}{4}$ for any dataset of size $n \geq n_0$. Then $n_0 = \Omega\left(\frac{\log^2(B)}{\alpha^2 \varepsilon^2 \log(1/\alpha)^4}\right)$.*

In particular, when $\varepsilon, \alpha = \Theta(1)$, the number of users must be $\Omega(\log^2 B)$ whereas our previous theorems show that $O(\log B)$ suffices for adaptive protocols. The authors believe that the high logarithmic dependence on $1/\alpha$ is an artifact of the proof.

**The Shuffle Model**   Shuffle differential privacy (shuffle-DP) (Bittau et al., 2017; Cheu et al., 2019) captures the idea that using a random permutation to shuffle a large enough set of somewhat private user messages, thus making their origins indistinguishable, boosts the privacy guarantee for each user. More precisely, in shuffle-DP, each user applies a LDP protocol to their data and then sends the output to a trusted *shuffler* whose only task is to randomly permute the users data before forwarding it to a central data curator. The privacy boost achieved from shuffling was analysed in (Feldman et al., 2021) (see Theorem G.4). To gain the privacy boost, the batch of users shuffled can not be too small. This makes it fundamentally incompatible with highly adaptive protocols having $n$ rounds of adaptivity, and each batch of size one. To bypass this, we consider protocols which run in a bounded number of rounds, shuffling the users queried in each round, simultaneously obtaining both the benefits of adaptivity and the boosted privacy from shuffling.

We refer to the problem of estimating the median of $n$ users within accuracy $\alpha$ using $r$ adaptive rounds of shuffling under $(\varepsilon, \delta)$-DP as shuffle-emp-median($\{x_i\}_{i=1}^{n}, \alpha, \varepsilon, \delta, r$) (see Section 2 for a formal definition). We provide a protocol for this problem with $r = \log_2 B$ and $n = (\log B) \cdot \widetilde{O}\left(\frac{1}{\varepsilon^2} + \frac{1}{\alpha^2}\right)$.

**Theorem 1.4.** *Let $r = \log_2 B$ and $\varepsilon, \alpha < 1$. There exists a protocol for* shuffle-emp-median($\{x_i\}_{i=1}^{n}, \alpha, \varepsilon, \delta, r$) *in the sequentially interactive model which requires*

$$n = O\left(\left(\frac{1}{\alpha^2} + \frac{1}{\varepsilon^2}\right) \log B \sqrt{\log(1/\delta) \log \frac{\log B}{\beta}}\right)$$

*users, and succeeds with probability $1 - \beta$. The protocol queries shuffled batches of $n/\log_2(B)$ users.*

We believe that the framework of combining shuffling with rounds of adaptivity might be of interest for many other problems. On the one hand, we could use a non-adaptive protocol with shuffling, getting better dependence on $\varepsilon$ and $\alpha$, but this would incur additional logarithmic factors in $B$. On the other hand, we could use a sequentially adaptive algorithm like in Theorem 1.1, but then we lose the benefits of shuffling since each batch has size 1. Theorem 1.4 demonstrates that protocols having several adaptive rounds using shuffling of each batch, can provide the best of both worlds.

**Experiments**   In Section 5, we demonstrate that the algorithm in Theorem 1.1 performs favorably compared to known non-adaptive mechanisms as well as a more naive noisy binary search mechanism.

## 1.1. Related Work

**Differential Privacy**   Differential privacy is considered the gold standard in private data analysis due to its rigorous guarantees, e.g., immunity to side information, and other useful properties (Dwork et al., 2006; Dwork & Roth, 2014). A number of mechanisms exist for releasing medians and general quantiles for centralized DP. First, one may instantiate mechanisms based on local sensitivity (Nissim et al., 2007; Dwork & Lei, 2009; Asi & Duchi, 2020), since quantiles often have low local sensitivity for many datasets. More recently, specialized mechanisms for medians (Tzamos et al., 2020; Drechsler et al., 2022; Aliakbarpour et al., 2024) and quantiles (Wilson et al., 2020; Gillenwater et al., 2021; Alabi et al., 2023) have been proposed to obtain even lower error but they require certain mild assumptions on the distribution of the data points. The case where data points can be arbitrary from some discrete domain $[B]$, like for us, has been well studied in the central setting. The sequence of works, (Beimel et al., 2016; Bun et al., 2015; Kaplan et al., 2020; Cohen et al., 2023) gradually reduced the number of users needed for accuracy $\alpha$ to $\tilde{O}\left(\frac{1}{\alpha \varepsilon} \log^*(B) \log^2(1/\delta)\right)$ for $(\varepsilon, \delta)$-privacy. This almost matches the the $\Omega(\log^*(B))$ lower bound from (Alon et al., 2019). A corollary of this lower bound is that even with central DP, no algorithm can achieve $o(1)$ quantile error in the continuous setting regardless of how many users there are.

**Local Differential Privacy**   There is increasing interest in local differential privacy (LDP), where the central aggregator is not trusted, and each user applies a DP mechanism to their data before broadcasting it. LDP mechanisms for many problems and accompanying lower bounds were shown in (Duchi et al., 2013). A ubiquitous LDP protocol that we will utilize is *randomized response* (See Lemma A.1), where answers to a binary query are flipped with probability $\frac{1}{1+e^\varepsilon} \approx \frac{1}{2} - \varepsilon$. For the median problem, an LDP algorithm was found in (Duchi et al., 2018) under a different loss function, the difference between the estimate and the median in the *data domain*. This loss function is subject to strong lower bounds (a linear dependence on the domain size). The most relevant work to our setting is the so-called *hierarchical mechanism* (Cormode et al., 2019).

**Hierarchical Mechanism**   The hierarchical mechanism uses the $b$-adic decomposition of the interval $[0, B]$ (which is a $b$-ary tree of depth $\Theta(\log_b(B))$ whose nodes at level $\ell$ correspond to intervals of length $\frac{B}{b^\ell}$). Each participant uniformly selects a level $\ell$ at random and employs standard frequency LDP oracles (Bassily & Smith, 2015; Wang et al., 2017) to disclose which node at level $\ell$ their data belongs to. The central aggregator may then combine the frequency oracles at each level to answer any range query. A particular use of range queries with relative error $\alpha$ is for

constructing an $\alpha$-approximate CDF of the data set, which in turn can be used to approximate every quantile within error $\alpha$. Unfortunately, dividing the user data among levels worsens the dependence on $\log(B)$. In Appendix D, we demonstrate that the hierarchical mechanism can be used to solve LDPstat-median with $n = O(\frac{\log^3 B}{\varepsilon^2 \alpha^2})$ users. In terms of the polynomial dependence on $\log B$, there is still a multiplicative $\log B$ gap between this upper bound and the lower bound of Theorem 1.3 which would be very interesting to close.

**Shuffle Differential Privacy**   The central model of DP requires that data be collected non-privately by the curator, which results in extremely accurate protocols. On the other hand, in the local model users do not trust anyone, and the response to any query must be privatized before it is broadcast by the user. In shuffle-DP, each user applies a LDP protocol to their data and then sends the output to a trusted *shuffler* whose only task is to randomly permute the users data before forwarding it to a central data curator. This places shuffling as a middle ground between these two models in terms of both trust and accuracy.

Understanding the separation between the local, shuffle, and central models of privacy, and therefore the trade-offs between trust and accuracy, is of both theoretical, and practical interest. For a survey of such separations, see (Cheu, 2021).

**Noisy Binary Search and Threshold Queries**   Consider an algorithm that sequentially picks a *threshold query* $m \in [B]$, then samples a user from the database $X$ and receives the bit $y = [x \le m]$. Since $\Pr[y = 1] = F_X(m)$, finding an integer $m$ such that $F_X(m) \approx q$ reduces to the *noisy binary search* problem. This search over a CDF with *threshold query* sample access, exactly mirrors searching over a monotonically increase sequence of coins. Noisy binary search was introduced by (Karp & Kleinberg, 2007) with a tight bound of $\Theta(\log(B)/\alpha^2)$, later improved by constant factors by (Gretta & Price, 2024), which holds for the non-private median when samples are accessed via threshold queries in the statistical setting.

**Structure of the Paper**   In Section 2, we introduce necessary preliminaries for our theoretical analyses. In Section 3, we provide an overview of our main ideas and technical contributions. Section 4 is dedicated to proving Theorem 1.1. In Appendix F, we prove the lower bounds of Theorems 1.2 and 1.3. In Appendix G, we provide the proof of Theorem 1.4. Finally, in Section 5 we present our experimental results.

## 2. Preliminaries

In local differential privacy, we assume that each of $n$ users hold a data point $x$ in the discrete and ordered domain $[B] = \{1, 2, \ldots, B\}$ for a positive integer $B$. Each user will communicate to a central (untrusted) aggregator using a differentially private mechanism. We consider sequentially adaptive protocols: In round $t$ the aggregator *queries* a set $I_t \subseteq \{1, \ldots, n\}$ of one or more parties, asking them to run a differentially private mechanism $\mathcal{M}_t$ on their data. The output of $\mathcal{M}_t(x_i)$ is then sent to the aggregator for each $i \in I_t$. In general, any sequentially adaptive protocol may be implemented by querying one new user over $n$ rounds[1]. Let us label the users $1, \ldots, n$ in the order in which the protocol queries them and denote the data of user $i$ by $x_i \in [B]$. Also denote the the private mechanism that user $i$ uses by $\mathcal{M}_i$ and the output $y_i = \mathcal{M}_i(x_i)$

Given the outputs $\{y_i\}_{i=1}^n$ where $y_i = \mathcal{M}_i(x_i)$, the data aggregator makes an estimate of the $q$th quantile with a post-processing function $\mathcal{F}$:

$$\tilde{m}_q = \mathcal{F}(y_1, \ldots, y_n).$$

We require that each $\mathcal{M}_i$ satisfy local differential privacy:

**Definition 2.1.** We say $\mathcal{M}_i$ satisfies $(\varepsilon, \delta)$-local DP if for all $x, x' \in [B]$, and all outputs $y$, we have

$$\Pr[\mathcal{M}_i(x) = y] \le e^{\varepsilon} \Pr[\mathcal{M}_i(x') = y].$$

We say that $\mathcal{M}_i$ satisfies $\varepsilon$-local DP if it satisfies $(\varepsilon, 0)$-DP.

In the adaptive setting, we allow $\mathcal{M}_i$ to depend on $y_1, \ldots, y_{i-1}$; i.e.

$$y_i = \mathcal{M}_i(x_i, y_1, \ldots, y_{i-1}). \tag{2}$$

where each $\mathcal{M}_i$ satisfies Definition 2.1 in $x_i$ (for any fixed choice of $y_1, \ldots, y_{i-1}$). In contrast, in a *non-adaptive* protocol, each $\mathcal{M}_i$ is fixed in advance (and usually all $\mathcal{M}_i$ are the same mechanism).

To measure the utility of $\tilde{m}_q$, we use the *quantile error*. For a given data set $X = (x_i)_{i=1}^n$ define $F_X$ as in Equation 1. We say $\tilde{m}_q$ is an $\alpha$-approximate quantile estimate on $X$ if

$$\Pr[[F_X(\tilde{m}_q), F_X(\tilde{m}_q + 1)] \cap (q - \alpha, q + \alpha) \ne \emptyset] \ge 1 - \beta,$$

where the above probability is over the randomness in $\tilde{m}_q$. We are typically interested in the high-probability setting, where $\beta = \frac{1}{\text{poly}(B)}$.

Now, we formally define the LDP median problems in both the statistical and empirical settings:

---

[1] Rounds which query multiple new users may be split into many rounds, each querying one user.

**Definition 2.2** (LDPstat-median). In $LDPstat\text{-}median(\mathcal{D}, n, \alpha, \varepsilon)$, $\mathcal{D}$ is an unknown distribution over $[B]$. Users $1, \ldots, n$ sample $x_1, \ldots, x_n$ according to $\mathcal{D}$. Each user $i$ outputs $y_i = \mathcal{M}_i(x_i, y_1, \ldots, y_{i-1})$ where the $\mathcal{M}_i$'s are $\varepsilon$-LDP mechanisms. The goal is to output an $\tilde{m} = \tilde{m}(y_1, \ldots, y_n) \in [B]$ such that $\tilde{m}$ is an $\alpha$-approximate median of $\mathcal{D}$.

**Definition 2.3** (LDPemp-median). In $LDPemp\text{-}median(\{x_i\}_{i=1}^n, \alpha, \varepsilon)$, there are users $1, \ldots, n$ (where the ordering is chosen by the protocol) with data points $(x_i)_{i=1}^n \in [B]^n$. User $i$ outputs $y_i = \mathcal{M}_i(x_i, y_1, \ldots, y_{i-1})$ where the $\mathcal{M}_i$'s are $\varepsilon$-LDP mechanisms. The goal is to output an $\tilde{m} = \tilde{m}(y_1, \ldots, y_n) \in [B]$ such that $\tilde{m}$ is an $\alpha$-approximate empirical median of $\{x_i\}_{i=1}^n$.

For shuffle DP, we assume that the protocol partitions the users $\{1, \ldots, n\}$ into $r$ disjoint subsets $I_1, \ldots, I_r$ and that each user $i \in I_t$ applies the same mechanism $\mathcal{M}_t$ to their data $x_i$ where $\mathcal{M}_t$ may be chosen adaptively based on $(\mathcal{M}_j(x_i))_{1 \leq j \leq t-1, i \in I_j}$. We assume that $\pi_t : I_t \to I_t$ is a uniformly random permutation for each $t \in [r]$. Given the outputs $(y_t)_{t \in [r]}$ where, $y_t = (\mathcal{M}_t(x_{\pi_t(i)}))_{i \in I_t}$ (in shuffled order), the data aggregator outputs

$$\tilde{m}_q = \mathcal{F}(y_1, \ldots, y_t),$$

for a post-processing function $\mathcal{F}$. We say that the protocol satisfies $(\varepsilon, \delta)$-shuffle DP if for any $t \in [r]$, any $(x_i)_{i \in I_t}, (x_i')_{i \in I_t}$ differing only in a single coordinate, and any set $S$,

$$\Pr[(\mathcal{M}_t(x_{\pi_t(i)}))_{i \in I_t} \in S]$$
$$\leq e^\varepsilon \Pr[(\mathcal{M}_t(x'_{\pi_t(i)}))_{i \in I_t} \in S] + \delta.$$

**Definition 2.4** (shuffle-emp-median). In $shuffle\text{-}emp\text{-}median(\{x_i\}_{i=1}^n, \alpha, \varepsilon, \delta, r)$, there are users $1, \ldots, n$ (where the ordering is chosen by the protocol) with data points $(x_i)_{i=1}^n \in [B]^n$. Using an $(\varepsilon, \delta)$-shuffle DP mechanism with $r$ rounds of adaptivity, the goal is to output an $\tilde{m} = \tilde{m}(y_1, \ldots, y_r) \in [B]$ such that $\tilde{m}$ is an $\alpha$-approximate empirical median of $\{x_i\}_{i=1}^n$.

## 3. Technical Contribution

In this section we give a high-level discussion our technical contribution for designing algorithms and proving lower bounds. For simplicity, we focus on the high privacy setting $\varepsilon \leq 1$.

### 3.1. Adaptive LDP Median Estimation via Noisy Binary Search (Theorem 1.1)

At the heart of our LDP median protocol of Theorem 1.1 is an algorithm for the noisy binary search problem from (Karp

& Kleinberg, 2007): Given an ordered set of $B$ coins with unknown head probabilities $\{p_i\}_{i=1}^B$ such that $p_1 \leq \cdots \leq p_B$, a target $\tau \in (0, 1)$, and an error $\alpha > 0$, our goal is to find any coin $i$ such that

$$[p_i, p_{i+1}] \cap (\tau - \alpha, \tau + \alpha) \neq \emptyset, \tag{3}$$

which intuitively means that the desired probability $\tau$ lies between coin $i$ and $i + 1$ (up to error $\alpha$). We refer to a coin satisfying the above property as $(\tau, \alpha)$-*good*. At each round, we may query a coin with index $j$, and we receive the result of the flipped coin. This problem generalizes classic binary search, where for the query $t$, one would have $p_i = 0$ for all $i \leq t$ and $p_i = 1$ for all $i > t$. We will denote the general problem as $\text{MonotonicNBS}(\{p_i\}_{i=1}^n, \tau, \alpha)$ (omitting $\{p_i\}_{i=1}^n$ when they are clear from context). The state-of-the-art algorithm for $\text{MonotonicNBS}$ is the *Bayesian Screening Search* (BayeSS) due to (Gretta & Price, 2024). Their algorithm finds a $(\tau, \alpha)$-good coin using $O(\frac{\log B}{\alpha^2})$ samples with high probability in $B$ [2].

To see how noisy binary search algorithms relate to median estimation under LDP, it is instructive to consider $\text{LDPstat-median}(\mathcal{D}, n, \alpha, \varepsilon)$. Concretely, any sample $x$ from $\mathcal{D}$ gives a coin flip with head probability $p_i = \Pr_{x \sim \mathcal{D}}[x \leq i]$ for any $i \in [B]$. It is a useful warmup problem, to show that one can solve $\text{LDPstat-median}$ using an algorithm for $\text{MonotonicNBS}$. Plugging in the algorithm of Gretta and Price gives an algorithm for $\text{LDPstat-median}(\mathcal{D}, n, \alpha, \varepsilon)$ if $n \geq C \frac{\log B}{\varepsilon^2 \alpha^2}$ for a constant $C$. We show the precise details in Appendix C.

For $\text{LDPemp-median}$, the situation is more complicated. A first idea is to reduce to the statistical setting by sampling users with replacement, thus sampling i.i.d from the empirical distribution. However, in sequentially adaptive protocols, users may only be queried once but sampling with replacement may sample a single user many times[3]. To resolve this issue, our main idea is to go through the users in a *random order* or equivalently sample users *without* replacement. Ideally, we would like to maintain the guarantees of algorithms for $\text{MonotonicNBS}$, but this problem assumes that the coin probabilities are unchanging over time. However,

---

[2]In fact, they obtain stronger guarantees. For any $\alpha, \tau$, their algorithm uses $\frac{1}{C_{\tau, \alpha}} \left( \log B + O(\log^{2/3} B \log^{1/3}(\frac{1}{\delta}) + \log(\frac{1}{\delta})) \right)$ where $C_{\tau, \alpha} = \Theta\left( \frac{\alpha^2}{\tau(1-\tau)} \right)$ for sufficiently small $\alpha$. Moreover, by information theoretic lower bounds, any algorithm must use $\frac{1}{C_{\tau, \alpha}} \log B$ coin flips.

[3]If we allow for multiple queries to the same user, we can indeed reduce to the statistical setting by sampling users with replacement. However, some users would then be sampled up to $O(\log n / \log \log n)$ times and to maintain $\varepsilon$-LDP, their reports would have to be made more noisy, thereby increasing the number of users needed to get an $\alpha$-approximate median. Thus, even allowing for users to be queried multiple times, it is unclear how to get optimal bounds via algorithms for $\text{MonotonicNBS}$.

when sampling without replacement, the empirical CDF of the remaining users, and thus the coin probabilities, change over time. Our main technical contribution is two-fold. We first show that throughout the process, no coin probability is altered too much.

**Lemma 3.1.** *Let $x_1, \ldots, x_n \in [B]$ and let $y_i = x_{\pi(i)}$ where $\pi : [n] \to [n]$ is a random permutation. For $0 \leq t < n$ and $j \in [B]$, we define $p_j^t = \frac{|\{t < i \leq n | y_i \leq j\}|}{n-t}$. Suppose that $n \geq C \frac{\log B}{\alpha^2}$ for a sufficiently large constant $C$. Then with high probability in $B$, we have for all $0 \leq t \leq n/2$ and all $j \in [B]$ that $|p_j^t - p_j^0| \leq \alpha$.*

Then, we show that the algorithm by Gretta and Price in fact also solves an *adversarial* version of `MonotonicNBS` which we denote `AdvMonotonicNBS`. Here, in each round, if coin $j$ is selected to be flipped, an adversary may instead flip a coin with a bias $p$ such that $|p_j - p| \leq c\alpha$ for some $c$. The goal is to return a $(\tau, \alpha(1+c))$-good coin. A formal definition can be found in Definition E.1. Our result, which may be of independent interest is as follows.

**Theorem 3.2.** *Let $0 < \alpha \leq \frac{1}{4}$ and suppose $c \leq 1$. There exists an algorithm for `AdvMonotonicNBS`$(1/2, \alpha, c)$ which uses $O(\frac{\log B}{\alpha^2})$ coin flips and returns an $(1/2, \alpha(1 + c))$-good with high probability in $B$.*

Now by Lemma 3.1, as we sample users without replacement, the CDF of the remaining users never changes by more than $\alpha$ at any point. In particular, for the data $x_j$ of a newly sampled user and a threshold $t \in [B]$, the probability of observing a one when applying randomized response to $[x_j \leq i]$ never varies by more than $\alpha\varepsilon$. Denoting this probability $p_i$, we are exactly in a position to apply Theorem 3.2 to conclude that $O(\frac{\log B}{\alpha^2\varepsilon^2})$ users suffices to find a $(1/2, 2\alpha\varepsilon)$ good coin. But this translates exactly to $i$ being an $O(\alpha)$-approximate median.

The proof of Lemma 3.1 and Theorem 3.2 can be found in Section 4 and Appendix E.

### 3.2. Lower bounds for Adaptive and Non-Adaptive Median Estimation (Theorems 1.2 and 1.3)

We next describe the main ideas for the lower bounds of Theorems 1.2 and 1.3, full proofs can be found in Appendix F.

**Lower Bound for Adaptive Protocols (Theorem 1.2)** In fact, we provide a lower bound for the general quantile estimation problem, demonstrating that all quantiles (not too close to the 0 or 1) are as hard as the median. To prove this lower bound, we first prove a lower bound in the statistical setting of Definition 2.2 and then reduce to the empirical setting of Definition 2.3. Our building block for the statistical lower bound is the framework in (Duchi et al., 2013), which uses the fact that a protocol attaining low error on the quantile problem, can distinguish distributions with

different $q$th quantiles from each other, even from a "hard" family of distributions. Our hard family of distributions will be close in statistical distance, but still has different $q$th quantiles:

$$
P_\beta(i) = \begin{cases} q - 2\alpha & i = 1 \\ 4\alpha & i = \beta \\ 1 - q - 2\alpha & i = B, \end{cases}
$$

for $\beta \in \{2, \ldots, B-2\}$. If $\beta$ is chosen uniformly at random, then our LDP distinguishing mechanism will be able to deliver $\log(B)$ bits of information (measured with the mutual information), by Fano's inequality. However, there is an upper bound on the amount of mutual information possible with an LDP protocol, as first established in (Duchi et al., 2013) and this leads to our desired result.

To get a lower bound in the empirical setting, we observe that a low-error algorithm for empirical quantile estimation can be applied to also get low-error in the statistical setting by just applying it on the data sampled from $\mathcal{D}$. The approximation guarantee follows from the fact that we have enough users that the empirical $q$-quantile of the samples is an $\alpha/2$ approximation to the true $q$-quantile of the distribution $\mathcal{D}$.

**Lower Bound for Non-Adaptive Protocols (Theorem 1.3)** It turns out more challenging to obtain a lower bound for non-interactive protocols. Our proof is via a reduction to the problem of privately learning a CDF under non-interactive LDP with $\ell_\infty$-error below $\alpha$. For small $\varepsilon$ and $\alpha$, it is known (Edmonds et al., 2020) that any such algorithm requires $\Omega(\frac{\log^2 B}{\varepsilon^2\alpha^2 \log^2(1/\alpha)})$ users[4].

Our reduction works as follows: Given a non-interactive $\varepsilon$-LDP algorithm for median estimation which succeeds with probability $2/3$, we first boost this success probability to $1 - \alpha^2$ with $O(\log 1/\alpha)$ independent repetitions and the median trick. The privacy of this protocol is thus $\varepsilon_1 = O(\varepsilon \log(1/\alpha))$. Second, assuming access to such an algorithm succeeding with high probability, we design a non-interactive CDF approximation algorithm as follows. First, we add $2n$ dummy users $n$ of which are 0 and $n$ of which are 1. We run the LDP median estimation algorithm on this new set of users and by selecting how many dummy users to include from the left and from the right, we can use their responses to estimate any quantile with error probability $\alpha_1 = O(\alpha)$ with probability $1 - O(\alpha^2)$. Union bounding over the equally spread $O(1/\alpha)$ quantiles $\alpha, 2\alpha, \ldots, \lfloor 1/\alpha \rfloor \cdot \alpha$, we obtain a CDF estimation algorithm which has error $\alpha_1$ with probability $1 - O(\alpha)$. In particular, the expected error of this non-interactive protocol is $O(\alpha)$. Now the lower bound from (Edmonds et al., 2020) kicks in

---

[4]In fact, their bound is $\Omega(\log^2 B)$, but it is relatively simple to check that their proof extends to general $\varepsilon, \alpha \leq 1$ with mild assumptions on these parameters.

which in turn gives the lower bound for median estimation, with the $\log^4(1/\alpha)$ stemming from the fact that we have to apply their lower bound with $\varepsilon_1 = O(\varepsilon \log(1/\alpha))$.

### 3.3. Shuffle DP for Median Estimation (Theorem 1.4)

Our core contribution with Theorem 1.4 is to demonstrate explicit trade-offs that exist when considering trust models and rounds of adaptivity. While adaptive algorithms that query $O(1)$ users per round are extremely sample efficient, they remain fundamentally incompatible with the shuffle model. We introduce protocols that exchange the benefits of faster learning for larger groups amenable to shuffling, and show that such protocols can compete in practical parameter regimes.

Karp & Kleinberg (2007) introduced a naïve "binary search with repetitions" algorithm as a baseline. This algorithm follows the natural approach of a standard binary search: query the midpoint, then recurse on either the left or right half of the array. However, since queries are noisy, each point must be queried a logarithmic number of times to achieve accuracy $\alpha$, and failure probability $\beta'$.

Building on the "near-optimal" analysis of (Feldman et al., 2021), we prove a shuffle-DP upper bound in Theorem 1.4 on the sample complexity of this "binary search with repetitions" algorithm with only $r = \log_2 B$ rounds of adaptivity. Sampling batches of $n/r$ users at each round we shuffle their private outputs, to learn one of the $r$ pivots up to accuracy $\alpha$ and failure probability $\beta/r$. Union bounding over all $r$ steps ensures we return an $\alpha$-approximate quantile with probability $1 - \beta$.

The full proof of Theorem 1.4 can be found in Appendix G.

## 4. Median Estimation with Adaptive LDP

In this section we prove Lemma 3.1 and Theorem 1.1, postponing the proof of Theorem 3.2 to Appendix E. We start with the following technical lemma.

**Lemma 4.1.** *Let $b_1, \ldots, b_{2n} \in \{0, 1\}$, $\pi : \{1, \ldots, 2n\} \to \{1, \ldots, 2n\}$ a random permutation, and $c_i = b_{\pi(i)}$ for $1 \leq i \leq 2n$. Let $Y_i = |\{j \in [2n] \setminus [i] \mid c_j = 0\}|$, and $X_i = \frac{Y_i}{2n-i} - \frac{Y_0}{2n}$. For any $t \geq 0$,*

$$\Pr\left[\max_{1 \leq i \leq n} |X_i| \geq t\right] \leq 2 \exp\left(\frac{-t^2 n}{2}\right).$$

*Proof.* We first note that $(X_i)_{i=0}^n$ forms a martingale. To see this, first observe that

$$\mathbb{E}[Y_{i+1} \mid (X_j)_{j \leq i}] = Y_i - \frac{Y_i}{2n-i}.$$

Indeed, conditioning on $(X_j)_{j \leq i}$, the probability that

$c_{i+1} = b_{\pi(i+1)} = 0$ is exactly $\frac{Y_i}{2n-i}$. Thus,

$$\mathbb{E}[X_{i+1} \mid (X_j)_{j \leq i}] = \frac{1}{2n-i-1}\left(Y_i - \frac{Y_i}{2n-i}\right) - \frac{Y_0}{2n}$$
$$= \frac{Y_i}{2n-i} - \frac{Y_0}{2n} = X_i$$

Moreover, writing $Y_{i+1} = Y_i - b$ where $b \in \{0, 1\}$ for a given $i < n$, we have

$$|X_{i+1} - X_i| = \frac{Y_i}{(2n-i)(2n-i-1)},$$

if $b = 0$, and

$$|X_{i+1} - X_i| = \frac{2n-i-Y_i}{(2n-i)(2n-i-1)},$$

if $b = 1$. Now, $Y_i$ is exactly the number of zeros among the $2n - i$ values $\pi(i+1), \ldots, \pi(2n)$, so trivially $0 \leq Y_i \leq 2n - i$. It follows that for $i < n$, in either of the cases $b \in \{0, 1\}$,

$$|X_{i+1} - X_i| \leq \frac{1}{2n-i-1} \leq \frac{1}{n}.$$

Finally, $X_0 = 0$, so we may apply Azuma's inequality (Theorem A.2 of Appendix A) with an appropriate rescaling of the $X_i$'s to obtain that

$$\Pr\left[\max_{1 \leq i \leq n} |X_i| \geq t\right] \leq 2 \exp\left(\frac{-t^2 n}{2}\right),$$

as desired. $\square$

It is now easy to obtain Lemma 3.1.

*Proof of Lemma 3.1.* Suppose without loss of generality that $n = 2n'$ is even. Fix $j \in B$ and define $b_i = [x_i \leq j]$ and $c_i = b_{\pi(i)} = [y_i \leq j]$ for $i \in [n]$. Let $Y_t = |\{t < i \leq 2n \mid c_i = 0\}|$. Then $p_j^t = \frac{Y_t}{n-t}$, so plugging into Lemma 4.1, we find that,

$$\Pr\left[\max_{0 \leq t \leq n'} |p_j^t - p_j^0| \geq \alpha\right] \leq 2 \exp\left(\frac{-\alpha^2 n}{2}\right)$$
$$\leq 2 \exp\left(\frac{-C \log B}{2}\right) \leq 2B^{-C/2}.$$

Choosing $C$ sufficiently large and union bounding over all $j \in [B]$, the result follows. $\square$

Finally, assuming Theorem 3.2, we can prove our main theorem Theorem 1.1.

*Proof of Theorem 1.1.* We pick a random permutation $\pi : [n] \to [n]$ and define $y_t = x_{\pi(t)}$, the input of user $\pi(t)$. For $j \in [B]$ and $t < n$, we define $q_j^t = \frac{|\{t < i \leq n | y_i \leq j\}|}{n-t}$ and

$q_0^t = 0$. Thus the map $j \mapsto q_j^t$ is the empirical CDF of the users $y_{t+1}, \ldots, y_n$.

Our algorithm uses the algorithm of Theorem 3.2 to solve AdvMonotonicNBS$(1/2, \alpha\varepsilon/8, 1)$ with the adversarial probabilities $\{p_j^t\}_{j=1}^B$ to be described shortly. To do so, whenever the algorithm calls for flipping a coin $j$ at step $t$, we sample a new user $x_{\pi(t)}$ and apply randomized response to the variable $[x_{\pi(t)} \leq j]$, retaining the bit with probability $\frac{e^\varepsilon}{1+e^\varepsilon}$ and flipping it otherwise, to get a variable $z_j^t$. By standard properties of randomized response, this protocol satisfies the $\varepsilon$-LDP requirement. Moreover, the probability $p_j^t$ that $z_j^t = 1$ is $p_j^t = q_j^t \cdot \frac{e^\varepsilon}{1+e^\varepsilon} + (1 - q_j^t) \cdot \frac{1}{1+e^\varepsilon}$ and so

$$|p_j^t - 1/2| = \left| \lambda_j^t \cdot \frac{e^\varepsilon - 1}{1 + e^\varepsilon} \right| \geq \frac{\varepsilon|\lambda_j^t|}{4}, \quad (4)$$

where we have written $q_j^t = 1/2 + \lambda_j^t$. Using that $n \gg \frac{\log B}{\varepsilon^2 \alpha^2} \gg \frac{\log B}{\alpha^2}$, it follows from Lemma 3.1, that $|q_j^t - q_j^0| \leq \alpha/5$ for all $t \leq n/2$ and $0 \leq j \leq B$ with high probability in $B$. Thus,

$$|p_j^t - p_j^0| = \frac{|q_j^t - q_j^0|(e^\varepsilon - 1)}{1 + e^\varepsilon} \leq \frac{\varepsilon\alpha}{10},$$

where the bound $\frac{e^\varepsilon - 1}{1+e^\varepsilon} \leq \varepsilon/2$ follows from a second degree Taylor expansion of the maps $f : \varepsilon \mapsto \frac{e^\varepsilon - 1}{1+e^\varepsilon}$ observing that $f'(0) = 1/2$ and $f''(\varepsilon) < 0$.

It now follows from Theorem 3.2, that using the noisy feedback from at most $n/2$ of the users, the algorithm finds an $(1/2, \frac{\alpha\varepsilon}{4})$-good coin $j^*$ with high probability in $B$. In particular $p_{j^*}^0 \leq \frac{1}{2} + \frac{\alpha\varepsilon}{4}$ and $p_{j^*+1}^0 \geq \frac{1}{2} - \frac{\alpha\varepsilon}{4}$. It thus follows from equation (4) that $q_{j^*}^0 \leq 1/2 + \alpha$ and $q_{j^*+1}^0 \geq 1/2 - \alpha$. Therefore $j^* + 1$ is an $\alpha$-approximate median of $\{x_i\}_{i=1}^n$ completing the proof. $\square$

# 5. Experiments

We compared three mechanisms for median estimation in the empirical setting: DpNaiveNBS (binary search with randomized response), Hierarchical Mechanism from (Cormode et al., 2019), which serves as the state of the art for non-adaptive protocols, and our sequentially adaptive algorithm, DpBayeSS, introduced in Theorem 1.1 (Algorithm 4 in Appendix I illustrates the pseudocode). Further details of the implementation, extensive experimental analysis, and experimental results for our algorithm in the shuffle model appear in Appendix I. Experiments were conducted on data generated from two distributions: a Pareto distribution over $[B]$, often used to model quantities like income and population (Arnold, 2014), and a uniform distribution over a random interval $[l, r]$ with $1 \leq l \leq r \leq B$, ensuring that the position of the median is not straightforward.

We evaluated the mechanisms using two metrics: the *success rate*, computed as the fraction of times a $(\frac{1}{2}, \alpha_{\text{test}})$-good

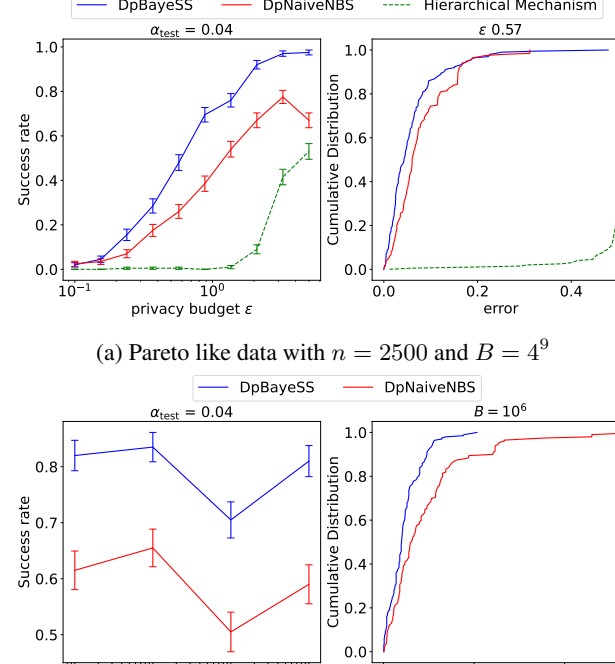

(a) Pareto like data with $n = 2500$ and $B = 4^9$

(b) Uniform data with $n = 2500$ and $\varepsilon = 1$

*Figure 1.* Plots 1a compare the three algorithms on the Pareto-like dataset: the left plot shows the success rate for $\alpha_{\text{test}} = 0.04$ across $\varepsilon \in [0.1, 5]$, and the right plot shows the c.d.f. of the absolute error for $\varepsilon = 0.57$. Plots 1b compare DpBayeSS and DpNaiveNBS on a uniform dataset with $\varepsilon = 1$: the left plot shows the success rate for different coin domains $B$ for $\alpha_{\text{test}} = 0.04$, and the right plot shows the c.d.f. of the absolute error for $B = 10^6$. The error bars on the left plots are standard deviation, computed as the sample average over 200 trials. The decrease in accuracy observed in Figure 1b at $B = 10^5$ is likely attributable to a random generation of a more challenging dataset.

coin is returned with $\alpha_{\text{test}} = 0.04$, and the *absolute quantile error*, $|F_X(\tilde{m}) - F_X(m)|$, where $\tilde{m}$ is the returned median. We run each algorithm 200 times and computed the standard deviation of the success rate as the sample average of a Bernoulli random variable.

In Figure 1a, we plot the success rate of the three privacy mechanisms for a fixed $B = 4^9$, and $n = 2500$, with $\varepsilon$ varying from 0.1 to 5, for the synthetic Pareto-like dataset. We also plot the cumulative distribution of the absolute quantile error, showing the distribution of the quantile error over 200 trials, for $\varepsilon = 0.57$. These parameter settings are typical values encountered in real applications; we test many more parameter values in Appendix I with similar results. The plots illustrate that DpBayeSS always achieves far higher success rate than the other two mechanisms, and is statistically significant as the confidence intervals are far from overlapping. Correspondingly, the CDF of the absolute quan-

tile error shows this value is much lower for `DpBayeSS`. Also, we observe that the number of users is insufficient to obtain a meaningful median using the `Hierarchical Mechanism`, which aligns with our theoretical predictions.

In Figure 1b, we plot the succes rate of `DpBayeSS` and `DpNaiveNBS` with a fixed privacy budget $\varepsilon = 1$ over varying domain sizes $B$ from $10^3$ to $10^6$ using the uniform distribution data set with 2500 users. We also plot the CDF of absolute quantile error for a large domain of $B = 10^6$. We observe again that `DpBayeSS` achieves superior performance in all the values of $B$ tested. Due to implementation constraints, `Hierarchical Mechanism` was not tested on this dataset, but results in Figure 1a indicate its error is generally higher than binary search-based methods. Our code is freely available [5].

## 6. Conclusion

Our starting observation is that approximating a quantile with limited information is essentially a noisy binary search. Based on this we introduced a novel adaptive local differentially private algorithm for estimating a quantile in both statistical and empirical settings. We have shown that this algorithm, which uses a private implementation of the Bayesian Screening Search of Gretta & Price (2024), is optimal up to constant factors. In proving this, we demonstrated that the original algorithm is robust to a limited form of adversarial noise addition, which may be of independent interest.

As adaptivity can be expensive, we also proposed a shuffle-DP mechanism that uses fewer rounds of communication by "batching" users and shuffling their private responses. We showed that this approach is competitive in practical parameter regimes and evaluated the effectiveness of these approaches through experiments on synthetic data designed to mimic real-world scenarios.

Several open questions remain. Absent privacy constraints in the statistical setting, one can estimate the median with $O(1/\alpha^2)$ samples (trivially, in the empirical setting, one simply outputs the median of the data set), so finding a protocol that converges to this as $\varepsilon \to \infty$ is of particular theoretical interest.

## Acknowledgements

Aamand, Pagh, and Imola carried out this work at Basic Algorithms Research Copenhagen (BARC), which was supported by the VILLUM Foundation grant 54451. Pagh and Imola were supported by a Data Science Distinguished Investigator grant from the Novo Nordisk Fonden. Boninsegna

was supported in part by the Big-Mobility project by the University of Padova under the Uni-Impresa call, by the MUR PRIN 20174LF3T8 AHeAD project, and by MUR PNRR CN00000013 National Center for HPC, Big Data and Quantum Computing.

## Impact Statement

This paper presents work whose goal is to advance the field of Private Machine Learning. There are many potential social consequences to collecting user data, even privately, none of which we feel must be specified here.

---

[5] https://github.com/NynsenFaber/Quantile_estimation_with_adaptive_LDP

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

# A. Additional Definitions

**Lemma A.1** (Binary Randomized Response (Warner, 1965; Dwork et al., 2006)). *For a binary input $x \in \{0, 1\}$, and privacy parameter $\varepsilon$, the following protocol $\mathcal{M} \to \{0, 1\}$ satisfies $\varepsilon$-LDP:*

$$\mathcal{M}(x) = \begin{cases} x, & \text{w.p. } \frac{e^\varepsilon}{e^\varepsilon + 1} \\ 1 - x, & \text{otherwise.} \end{cases}$$

**Azuma's inequality** We will use the following version of Azuma's inequality which bounds the maximum deviation of a martingale $(X_i)_{i=0}^n$ at any time $t = 0, \ldots, n$. See Theorem 2.1 in (Fan et al., 2012) for a stronger and more general bound.

**Theorem A.2** (Azuma's inequality). *Let $(X_i)_{i=0}^n$ be a martingale such that $X_0 = 0$ and $|X_{i+1} - X_i| \leq 1$ for all $0 \leq i < n$. For any $t \geq 0$,*

$$\Pr[\max_{1 \leq i \leq n} |X_i| \geq t] \leq 2 \exp\left(\frac{-t^2}{2n}\right).$$

**Bernstein's Inequality** We use the following variant of Bernstein's Inequality in the proof of Theorem 1.4, see Wainwright (2019, Proposition 2.10) for a detailed overview.

**Theorem A.3** (Bernstein's Inequality). *Let $\{X_i\}_{i=1}^n$ be independent random variables that are bounded almost surely by 1. Let $\sigma^2 = \frac{1}{n} \sum_{i=1}^n \text{Var}[X_i]$ be the average variance. We then have,*

$$\Pr\left[\left|\frac{1}{n} \sum_{i=1}^n X_i - \frac{1}{n} \sum_{i=1}^n \mathbb{E}[X_i]\right| > \alpha\right] \leq \exp\left(\frac{-n\alpha^2}{2\sigma^2 + \frac{2\alpha}{3}}\right).$$

# B. Reduction to the Median

Consider the simple case where we are given an algorithm $A$ which returns the median of $n$ samples in the most natural sense, by returning the $n/2$'th index of their sorted representation. Without changing this algorithm we can have it return any arbitrary index by adding elements to the beginning or the end of this sorted array. For example, adding two elements to the beginning of the array will create a new array with $n' = n + 2$ elements where the $n'/2$'th index will be the $(n/2 - 1)$'th index of the original array. The padding argument below formalizes this notion, demonstrating that any algorithm for an $\alpha$-approximation of the median can be used to obtain a $2\alpha$-approximation of any quantile.

**Lemma B.1** (Padding Argument). *Any $\alpha$-approximation algorithm for the median, with $\alpha \in \left(0, \frac{1}{2}\right)$, can be used to construct a $2\alpha$-approximation for any quantile $\tau \in (0, 1)$.*

*Proof.* Consider a dataset $D = \{x_1, \ldots, x_n\}$ where each element is such that $x_i \in \{1, \ldots, B\}$. Let $\mathcal{M}$ be an algorithm for the $\alpha$-approximation of the median then for $m = A(D)$ we have by definition

$$\Pr_{\mathcal{D}}[x \leq m] < \frac{1}{2} + \alpha \qquad \text{and} \qquad \Pr_{\mathcal{D}}[x \leq m + 1] > \frac{1}{2} - \alpha. \tag{5}$$

where $\Pr_D[x \leq m] = \frac{\sum_{x \in D}[x \leq m]}{n}$, and $[x \leq m]$ is an indicator function. Consider now a padded dataset $D_P = D \cup \{1\}^{(1-\tau)n} \cup \{B\}^{\tau n}$, where $\{a\}^x$ indicates the multi-set containing the $a$ element $x$ times [6]. The new empirical cumulative distribution of the data set for $y \in \{1, \ldots, B - 1\}$, is

$$\Pr_{D_P}[x \leq y] = \frac{(1 - \tau)n + \sum_{x \in D}[x \leq y]}{|D_P|} = \frac{1 - \tau}{2} + \frac{1}{2}\Pr_D[x \leq y],$$

as we have $|D_P| = 2n$. Thus

$$\Pr_D[x \leq y] = 2\Pr_{D_P}[x \leq y] + \tau - 1. \tag{6}$$

The application of $A$ to the padded data set $D_P$ returns a $\alpha$-approximate median $m_P = A(D_P)$. Therefore, for $m_P \in \{1, \ldots, B - 1\}$, from Equation 6 and Equation 5 it follows that

$$\Pr_D[x \leq m_P] < \tau + 2\alpha \qquad \text{and} \qquad \Pr_D[x \leq m_P + 1] > \tau - 2\alpha. \tag{7}$$

Notice that $m_p \neq B$, as $\Pr_{D_P}[x \leq B] = 1 < \frac{1}{2} + \alpha$ iff $\alpha > \frac{1}{2}$. This concludes the proof. $\square$

---

[6]We consider $(1 - \tau)n$ and $\tau n$ integers.

## C. Statistical Private Median Estimation

In this section, we will provide an algorithm for `LDPstat-median` using the state-of-the-art algorithm for `MonotonicNBS`. We prove the following:

**Theorem C.1.** *Let* $\alpha \in \left(0, \frac{1}{4}\right)$ *and* $\varepsilon > 0$. *Suppose that the number of users* $n \geq C \frac{\log B}{\alpha^2} \left(\frac{e^\varepsilon + 1}{e^\varepsilon - 1}\right)^2$ *for a sufficiently large constant* $C$. *Then there exists an algorithm solving* `LDPstat-median`$(\mathcal{D}, n, \alpha, \varepsilon)$ *with high probability in* $B$.

In this section, we prove Theorem C.1. For this, we recall the following result which is a corollary of the main result in (Gretta & Price, 2024). Recall the definition of an $\left(\frac{1}{2}, \alpha\right)$-good coin in (3).

**Theorem C.2** ((Gretta & Price, 2024)). *For any* $\alpha \in \left(0, \frac{1}{4}\right)$, *there exists an algorithm for* `MonotonicNBS`$(\tau, \alpha)$ *which uses* $O(\frac{\log B}{\alpha^2})$ *coin flips and outputs an* $\left(\frac{1}{2}, \alpha\right)$-*good coin with high probability in* $B$.

*Proof of Theorem C.1.* For $i \in [B]$, we define $q_i = \sum_{j \leq i} \mathcal{D}[j]$ with the convention that $q_0 = 0$. Thus $j \mapsto q_j$ is the CDF of $\mathcal{D}$. Consider sampling $X \sim \mathcal{D}$ and let $Y$ be the random variable obtained by applying randomized response to the indicator variable $[X \leq j]$ retaining the bit with probability $\frac{e^\varepsilon}{1+e^\varepsilon}$ and flipping it otherwise. Then $\Pr[Y = 1] = p_j$ where $p_j = q_j \cdot \frac{e^\varepsilon}{1+e^\varepsilon} + (1 - q_j) \cdot \frac{1}{1+e^\varepsilon}$. Then,

$$q_j = \left(p_j - \frac{1}{e^\varepsilon + 1}\right) \frac{e^\varepsilon + 1}{e^\varepsilon - 1}, \tag{8}$$

We use the the algorithm in Theorem C.2 to solve `MonotonicNBS`$\left(\frac{1}{2}, \alpha \frac{e^\varepsilon - 1}{e^\varepsilon + 1}\right)$ when the inputs are the unknown $\{p_i\}_{i=1}^B$. To do so, whenever the algorithm calls for flipping a coin $j$, we sample a new user $X \in \mathcal{D}$ and apply randomized response to the variable $Y = [X \leq j]$. By standard properties of randomized responze, this protocol satisfies the $\varepsilon$-LDP requirement. Moreover, by Theorem C.2, the algorithm finds an $\left(\frac{1}{2}, \alpha \frac{e^\varepsilon - 1}{e^\varepsilon + 1}\right)$-good coin $j^*$ with high probability in $B$. In particular $p_{j^*} \leq \frac{1}{2} + \alpha \frac{e^\varepsilon - 1}{e^\varepsilon + 1}$ and $p_{j^*+1} \geq \frac{1}{2} - \alpha \frac{e^\varepsilon - 1}{e^\varepsilon + 1}$. It thus follows from Equation (8) that $q_{j^*} \leq 1/2 + \alpha$ and $q_{j^*+1} \geq 1/2 - \alpha$. Therefore $j^*$ is an $\alpha$-approximate median of $\mathcal{D}$ completing the proof. $\square$

In the high privacy regime, i.e. for $\varepsilon < 1$, the sample complexity of Theorem C.1 becomes $n = \Omega\left(\frac{\log B}{\varepsilon^2 \alpha^2}\right)$, matching our lower bound up to a constant factor. 00

## D. The Hierarchical Mechanism

The algorithm was presented in (Cormode et al., 2019) and can be used to approximately answer general range queries. It comes in several variants but here will show how it can be used to reconstruct the CDF of a distribution $\mathcal{D}$ on $[B]$ within $\ell_\infty$-error $\alpha$ under $\varepsilon$-LDP in a non-adaptive fashion using $O(\frac{(\log B)^3}{\varepsilon^2 \alpha^2})$ users sampled i.i.d from $\mathcal{D}$ with high probability in $B$. In particular, this implies that it can solve `LDPstat-median` (in fact approximate any quantile) within the same sample complexity with high probability in $B$.

The main idea is to construct a $b$-ary tree of depth $\Theta(\log(B))$ on $[B]$. For the below, we will assume that $B$ is a power of 2 and that $b = 2$ (although for the experiments, we use a different constant $b$). The nodes on level $i$ (where level 0 is the root) corresponds to the $2^i$ dyadic intervals of $B$. Namely, in the binary representation of elements of $B$, there is an interval corresponding to each prefix of length $i$ in the binary representation. The non-adaptive protocol we will consider is as follows. The $n$ samples $x_1, \ldots, x_n \sim \mathcal{D}$ are partitioned into batches of size $k := n/(\log B + 1)$, one for each level $\ell = 0, \ldots, \log B$ of the binary tree. Namely, for $\ell = 0, \ldots, \log B$, define $S_\ell = (x_{k\ell+1}, \ldots, x_{k(\ell+1)})$. A user $i$ with $k\ell + 1 \leq i \leq k(\ell + 1)$ writes a one-hot encoding $z$ of which node they belong to on level $\ell$ and uses randomized response on each of the $2^\ell$ bits of $z$ retaining them with probability $\frac{e^\varepsilon}{1+e^\varepsilon}$ and flipping them with probability $\frac{1}{1+e^\varepsilon}$. This is the message $y$, they send to the central server. This is the unary encoding mechanism; see (Cormode et al., 2019) for more sophisticated solutions, that require less communication but nonetheless have the same approximation errors. The combined algorithm is denoted `Hierarchical Mechanisms`.

**Analysis sketch of `Hierarchical Mechanism`** We here analyse the performance of `Hierarchical Mechanism` for approximating the CDF $F_{\mathcal{D}}(i)$ of $\mathcal{D}$ at any point $i \in [B]$. We assume that $\varepsilon \leq 1$. Denote by $I = \{1, \ldots, i\}$ and let $I = I_1 \cup \cdots \cup I_t$ be the unique partition of $I$ into $t \leq \log B$ dyadic intervals such that for each $a$, at most one of

---

**Algorithm 1** `BayeSS` main steps

---

**Input:** $\{x_i\}_{i=1,\ldots,n}$, $\alpha \in (0, 1/4)$, $n \geq C\frac{\log B}{\alpha^2}$
$L \leftarrow \texttt{BayesLearn}(B, \{x_i\}_{i=1,\ldots,n/4}, \alpha)$
$R \leftarrow \frac{1}{\gamma}$-quantiles$(L)$ {for $\gamma = O(1)$}
**return** $\texttt{TestCoins}(R, \{x_i\}_{n/4+1,\ldots,n/2}, \alpha)$

---

the intervals have length $2^a$. For a sample $x \sim \mathcal{D}$, denote by $\lambda_j = \Pr[x \in I_j]$ such that $\lambda := \sum_{j=1}^{t} \lambda_j$ is exactly equal to $F_{\mathcal{D}}(i)$. Let $\ell_1, \ldots, \ell_t$ be the levels of the tree corresponding to $I_1, \ldots, I_t$. For $1 \leq j \leq t$, the bit corresponding to $I_j$ of the randomized response vectors obtained from $S_{\ell_j}$ constitute $k$ i.i.d random $\{0, 1\}$ variables $X_1^j, \ldots, X_k^j$ each attaining the value 1 with probability $\frac{\lambda_i e^\varepsilon}{1+e^\varepsilon} + \frac{(1-\lambda_i)}{1+e^\varepsilon}$. Namely, they can attain the value 1 exactly if the sample is in $I_j$ (which happens with probability $\lambda_j$) and the bit is retained, or if the sample is not in $I_j$ (which happens with probability $1 - \lambda_j$) but the bit is flipped. For $r = 1, \ldots, k$, define

$$Y_r^j = \frac{1}{k}\left(X_r^j - \frac{1}{1+e^\varepsilon}\right) \cdot \frac{e^\varepsilon + 1}{e^\varepsilon - 1},$$

so that $\mathbb{E}[Y_r^j] = \lambda_j/k$. Since $\varepsilon \leq 1$, we have $|Y_r^j| \leq C/(\varepsilon k)$ for some universal constant $C$. Define,

$$X = \sum_{j=1}^{t} \sum_{r=1}^{k} Y_r^j$$

which is a sum of $tk$ independent (but not necessarily i.i.d) random variables. Then $\mathbb{E}[X] = \lambda$, and it follows from Hoeffding's inequality that

$$\Pr[|X - \lambda| \geq \alpha] \leq 2\exp\left(-\frac{2\alpha^2}{kt(2C/(k\varepsilon))^2}\right) = 2\exp\left(-\frac{\alpha^2 \varepsilon^2 k}{2C^2 t}\right).$$

Since $t \leq \log B$, it follows that if $k = O((\log B)^2/(\varepsilon^2\alpha^2))$ is sufficiently large, then

$$\Pr[|X - \lambda| \geq \alpha] \leq B^{-\gamma}$$

for any desired constant $\gamma$. In particular, it suffices to have $k = O((\log B)^2/(\varepsilon^2\alpha^2))$ users at each level of the tree to approximate the CDF with high probability within $\alpha$ under $\varepsilon$-LDP. Since $n = k(\log B + 1)$, the claim on the sample complexity follows.

## E. Proof of Theorem 3.2

The goal of this section is to prove Theorem 3.2. We first define the adversarial setting.

**Definition E.1.** Let $0 < \alpha < 1$ and $B$ a positive integer. Let $p_0, \ldots, p_B \in [0, 1]$ be unknowns with $0 = p_0 \leq \cdots \leq p_B = 1$. In $AdvMonotonicNBS(\tau, \alpha, c)$, for $c > 0$, our goal is to identify an $(\tau, \alpha(1+c))$-good coin (defined in Equation 3). To do so, we may iteratively pick indices $i \in B$. Then an adversary selects a probability $\tilde{p}_i$ such that $|\tilde{p}_i - p_i| \leq c\alpha$, and we observe the outcome of a coin flip with heads probability $\tilde{p}_i$.

We show that the `BayeSS` algorithm (`BayeSS` abbreviates *Bayesian Screening Search*) from (Gretta & Price, 2024)(Algorithm 3) solves the $AdvMonotonicNBS(\tau, \alpha, c)$ problem returning the a $(\tau, \alpha(1+c))$-good coin with high probability in $B$ using $O(\frac{\tau(1-\tau)\log B}{\alpha^2})$ coin flips. We actually prove a stronger theorem which immediately implies Theorem 3.2.

**Theorem E.2.** *Suppose that $c \leq 1$ and $\alpha \leq \frac{1}{2}\min\{\tau, 1 - \tau\}$. There exists an algorithm (Gretta & Price, 2024) for $AdvMonotonicNBS(\tau, \alpha, c)$ which uses $\frac{1}{C_{\tau,\alpha}}(\log B + O(\log^{2/3} B \log^{1/3}\frac{1}{\beta} + \log\frac{1}{\beta}))$ coin flips[7] and returns a $(\tau, \alpha(1+c))$-good coin with probability at least $1 - \beta$.*

---

[7]Namely, $C_{\tau,\alpha}$ is the information capacity of the Binary Asymmetric Channel (BAC) with crossover probabilities $\{\tau + \alpha, \tau - \alpha\}$. Concretely, $C_{\tau,\alpha} = \max_q H((1-q)(\tau - \alpha) + q(\tau + \alpha)) - (1-q)H(\tau - \alpha) - qH(\tau + \alpha)$ with $H$ being the binary entropy function, and $C_{\tau,\alpha} = \Theta(\frac{\alpha^2}{\tau(1-\tau)})$ for $\alpha \leq \frac{1}{2}\min(\tau, 1 - \tau)$.

Note that Theorem 3.2 follows directly from Theorem E.2 by setting $\tau = 1/2$ and $\beta = B^{-\lambda}$ for any constant $\lambda$. With this, the proof of Theorem 1.1 is complete.

Before we delve into the proof of Theorem E.2, let us first describe the idea behind `BayeSS`, described shortly in Algorithm 1. At a high level `BayeSS` proceeds in two steps allocating a portion of the coin flips for each step. The first step is a Bayes learner algorithm, called `BayesLearn`. It starts by assigning a uniform prior $w(I_i)$ to each coin interval $I_i = [i, i+1]$ for any $i \in [B-1]$, then takes the $\tau$-quantile interval under the posterior $w(I_i)$, selects a coin from this interval, flips it, and updates each $w(I_i)$ according to the result of the coin flip and the error $\alpha$. This procedure is repeated iteratively. The sampled intervals are collected in a multiset $L$, with the guarantee that, after $O\left(\frac{(1+\gamma)\log B}{C_{\tau,\alpha}}\right)$ coin flips, a $\gamma$-fraction of intervals in $L$ contains a $(\tau, \alpha)$-good coin with high probability in $B$ (referred to as good intervals). In the second step, this property is used to narrow the set of possible coins to $O(1/\gamma)$, ensuring that it contains at least one $(\tau, \alpha)$-good coin. Each coin in the candidate set can be individually tested, up to error $\alpha$, with high probability using $O(\frac{1}{\gamma\alpha^2}\log(\frac{B}{\gamma}))$ coin flips.

It is easy to see that in the adversarial setting, the coins can be tested up to error $\alpha(1+c)$ in the second step. Our main challenge in proving Theorem E.2, is analyzing the first part of the algorithm, `BayesLearn`, in the adversarial setting. The authors in (Gretta & Price, 2024) used a stopping time argument to analyze `BayesLearn`. They defined a potential function $\Phi$, with an initial negative value, constructed so that a positive potential implies finding at least a $\gamma$ fraction of good intervals. The stochastic process describing the evolution of the potential $\{\Phi_i\}_{i=1,\dots}$ is then modeled with a submartingale that can be used to bound, using Azuma's inequality, the probability that the process crosses zero after a sufficient number of iterations. We prove that we can use the same argument for the case of adversarial probabilities if we allow the potential to catch approximate good intervals, namely intervals containing $(\tau, \alpha(1+c))$-good coin.

**New Potential**   Let $\{\ell, \dots, r\}$ be the set of $(\tau, \alpha(1+c))$-good intervals. Let $a$ be the maximum $i \in [B-1]$ such that $p_i^1 \leq \tau$. Let $L$ be the list of intervals visited in `BayesLearn`. We define the potential function as

$$\Phi(w, L) := \log_2 w(a) + 12 C_{\tau,\alpha}(|\{x \in L : x \in [\ell, r]\}| - \gamma|L|),$$

where $w(a)$ is the Bayesian posterior weight associated to the best interval $a$ and $C_{\tau,\alpha}$ is a concrete function of $\tau$ and $\alpha$. Notice that a positive potential implies $|\{x \in L | x \in [\ell, r]\}| > \gamma|L|$, hence indicating the presence of a $\gamma$ fraction $(\tau, \alpha(1+c))$-good intervals in $L$. The following Lemma generalises Lemma 7 of (Gretta & Price, 2024) and allows the construction of a submartingale.

**Lemma E.3** (Adaptation of Lemma 7 in (Gretta & Price, 2024) for adversarial probabilities). *For $c \leq 1$ and $\alpha \leq \frac{1}{2}\min\{\tau, 1-\tau\}$, the expected variation of the potential is*

$$\mathbb{E}[\Phi_{t+1} - \Phi_t | y_1, \dots, y_t] \geq (1 - 12\gamma)C_{\tau,\alpha}, \tag{9}$$

*where $(y_1, \dots, y_t)$ are the results of the coin toss up to $t+1$-th sample, and $C_{\tau,\alpha} = \Theta\left(\frac{\tau(1-\tau)}{\alpha^2}\right)$.*

*Proof.* The proof for the adversarial setting, which allows an adversary to alter the head coin probability at each iteration up to $c\alpha$, while preserving their order, closely resembles the proof of Lemma 7 in (Gretta & Price, 2024), which addresses the case of fixed coin probabilities. We will go through the steps of the proof highlighting the main differences. An implementation of `BayesLearn` for empirical quantile estimation, where each user is used at most once, can be found in Algorithm 2.

Let's define the capacity of the $(\tau, \alpha)$-BAC (Binary Asymmetric Channel) as

$$C_{\tau,\alpha} = \max_q H((1-q)(\tau-\alpha) + q(\tau+\alpha)) - (1-q)H(\tau-\alpha) - qH(\tau+\alpha),$$
$$q = \arg\max_x H((1-x)(\tau-\alpha) + x(\tau+\alpha)) - (1-x)H(\tau-\alpha) - xH(\tau+\alpha),$$

where $H(p)$ is the binary entropy. Let's define the multiplicative Bayes weights $d_{x,y} : \{0,1\} \times \{0,1\} \to \mathbb{R}$, they indicates the multiplicative effect of a flip resulting $x$ (1=Heads, 0=Tails) on the density of an interval on side $y$ (1=Right, 0=Left) of

---

**Algorithm 2** `BayesLearn` for empirical quantile estimation, from Algorithm 2 in (Gretta & Price, 2024)

---

1: **function** `GetIntervalFromQuantile`$(w, q)$
2:     **return** $\min i \in [B]$ s.t. $W(i) \geq q$ **with** $W(x) = \sum_{i \in \{1,\ldots,x\}} w(i)$
3: **end function**

4: **function** `RoundIntervalToCoin`$(i, w, q)$
5:     **return** $i$ **if** $\frac{q - W(i-1)}{w(i)} \leq q$ **else** $i+1$ **with** $W(x) = \sum_{i \in \{1,\ldots,x\}} w(i)$
6: **end function**

7: **function** `BayesLearn`$(\{x_i\}_{i=1,\ldots,n}, B, \tau, \alpha, M)$
8:     $w_1 \leftarrow \text{uniform}([B-1])$
9:     $q \leftarrow \arg\max_x H((1-x)(\tau - \alpha) + x(\tau + \varepsilon)) - (1-x)H(\tau - \alpha) - xH(\tau + \alpha)$
10:    $I \leftarrow \{\}$ {Multiset}
11:    **for** $i \in [M]$ **do**
12:       $j_i \leftarrow$ `GetIntervalFromQuantile`$(w_i, q)$
13:       $c_i \leftarrow$ `RoundIntervalToCoin`$(j_i, w_i, q)$ {Gets the coin from the selected interval}
14:       $L \leftarrow L \cup \{j_i\}$
15:       $x_i \sim \{x_k\}_{k=1,\ldots}$ {Sample a user}
16:       $\{x_k\}_{k=1,\ldots} \leftarrow \{x_k\}_{k=1,\ldots} \setminus \{x_i\}$ {Remove the user from the dataset}
17:       $y_i \leftarrow [x_i \leq c_i]$ {Flip the coin}
18:       $w_{i+1}(x) \leftarrow \begin{cases} w_i(x)d_{\tilde{y}_i,0} & \text{if } x \in \{1, \ldots, j_i - 1\} \\ d_{\tilde{y}_i,0}(q - W_i(j_i - 1)) + d_{\tilde{y}_i,1}(W_i(j_i) - 1) & \text{if } x = j_i \\ w_i(x)d_{\tilde{y}_i,1} & \text{if } x \in \{j_i + 1, \ldots, B - 1\} \end{cases}$
19:    **end for**
20:    **return** $L$ {Return a multiset of intervals}
21: **end function**

---

the flipped coin.

$$d_{0,0} = \frac{1 - \tau - \alpha}{1 - \tau - (2q-1)\alpha}$$

$$d_{0,1} = \frac{1 - \tau + \alpha}{1 - \tau - (2q-1)\alpha}$$

$$d_{1,0} = \frac{\tau + \alpha}{\tau + (2q-1)\alpha}$$

$$d_{1,1} = \frac{\tau - \alpha}{\tau + (2q-1)\alpha}.$$

We will mainly use the results from Lemma 9 in (Gretta & Price, 2024) that states that

$$C_{\tau,\alpha} = (\tau + \alpha)\log_2 d_{1,0} + (1 - \tau - \alpha)\log_2 d_{0,0}, \tag{10}$$

$$C_{\tau,\alpha} = (\tau - \alpha)\log_2 d_{1,1} + (1 - \tau + \alpha)\log_2 d_{0,1}, \tag{11}$$

with the fact that $d_{1,0} \geq d_{0,0}$ and $d_{1,1} \leq d_{0,1}$. Recall the potential function: let $\{\ell, \ldots, r\}$ be the set of $(\tau, \alpha(1+c))$-good intervals. Let $a$ be the maximum $i \in [B-1]$ such that $p_i^1 \leq \tau$. Let $L$ be the list of intervals visited in `BayesLearn`.

Let $j_t$ be the interval chosen at $t$-th round, and let $c_t$ be the index of the coin flipped. Let $p_{c_t}^t = p^t$ (we will discard the coin subscript) the probability of the selected coin at time $t$. We split the potential in two addend

$$12C_{\tau,\alpha}(|\{x \in L | x \in [\ell, r]\}| - \gamma|L|) \tag{12}$$

$$\log_2 w(a) \tag{13}$$

The main difference with the proof in (Gretta & Price, 2024) is that a good coin is defined on the initial probabilities $\{p_i^1\}_{i=1,\ldots,B}$, but at the $t$-th iteration we only have access to coin with probability $\{p_i^t\}_{i=1,\ldots,B}$. However, they are concentrated around $\alpha$, so $|p^t - p^1| \leq c\alpha$ for $c \leq 1$.

**Bad Queries:** Consider $j_t \notin [\ell, r]$. If $j_t > r$, then $p^1 \geq \tau + (1+c)\alpha$. As we have that $|p^t - p^1| \leq c\alpha$ we also have $p^t \geq p^1 - c\alpha \geq \tau + (1+c)\alpha - c\alpha = \tau + \alpha$. The expected change in the weights is

$$\mathbb{E}[\log_2 w_{t+1}(a) - \log_2 w_t(a)] = p^t \log_2 d_{1,0} + (1 - p^t) \log_2 d_{0,0} \geq C_{\tau,\alpha}.$$

Where the last inequality comes from the fact that the expression is minimized as $p^t = \tau + \alpha$, and Equation 10. Consider now $j_t < L$, then $p^1 \leq \tau - (1+c)\alpha$, which means $p^t \leq p^1 + c\alpha \leq \tau - (1+c)\alpha + c\alpha = \tau - \alpha$, then

$$\mathbb{E}[\log_2 w_{t+1}(a) - \log_2 w_t(a)] = p^t \log_2 d_{1,1} + (1 - p^t) \log_2 d_{0,1} \geq C_{\tau,\alpha},$$

where we reach the minimum $C_{\tau,\alpha}$ when $p^t = \tau - \alpha$, due to Equation 11. As $j_t \notin [\ell, r]$ the change in Equation 12 is $-\gamma \cdot 12 C_{\tau,\alpha}$. Therefore, on bad queries the expected change in $\Phi$ is at least $(1 - 12\gamma) C_{\tau,\alpha}$.

**Good Queries:** Let's consider the expected change in Equation 13 when $j_t \in [\ell, r]$. Consider the case where $j_t \neq a$, then the expected change is either

$$p^t \log_2 d_{1,0} + (1 - p^t) \log_2 d_{0,0} \quad \text{if} \quad a \text{ is on the left of } j_t, \text{ so } p^0 \geq \tau \Rightarrow p^t \geq \tau - c\alpha$$
$$p^t \log_2 d_{1,1} + (1 - p^t) \log_2 d_{0,1} \quad \text{if} \quad a \text{ is on the right of } j_t, \text{ so } p^0 \leq \tau \Rightarrow p^t \leq \tau + c\alpha$$

The first expression is increasing in $p^t$ while the second is decreasing, therefore the expected change is at least

$$\min\left\{(\tau - c\alpha) \log_2 d_{1,0} + (1 - \tau + c\alpha) \log_2 d_{0,0} \, ; \, (\tau + c\alpha) \log_2 d_{1,1} + (1 - \tau - c\alpha) \log_2 d_{0,1}\right\} \tag{14}$$

Let's consider the first argument of the previous expression

$$(\tau - c\alpha) \log_2 d_{1,0} + (1 - \tau + c\alpha) \log_2 d_{0,0} = (\tau + \alpha) \log_2 d_{1,0} + (1 - \tau - \alpha) \log_2 d_{0,0} - \alpha(1+c)(\log_2 d_{1,0} - \log_2 d_{0,0})$$
$$= C_{\tau,\alpha} - \alpha(1+c) \underbrace{(\log_2 d_{1,0} - \log_2 d_{0,0})}_{\geq 0} \qquad (\text{as } d_{1,0} \geq d_{0,0})$$
$$\geq C_{\tau,\alpha} - 2\alpha(\log_2 d_{1,0} - \log_2 d_{0,0}) \qquad (\text{as } c \leq 1)$$
$$\geq C_{\tau,\alpha} - 2(6 \log 2) C_{\tau,\alpha}$$
$$\geq -11 C_{\tau,\alpha},$$

where in the first inequality we used the fact that $c \leq 1 \Rightarrow (1+c)\alpha \leq 2\alpha$, while in the second inequality we used Lemma 10 and Lemma 13 in (Gretta & Price, 2024), valid for $\alpha \leq \frac{1}{2}\min(\tau, 1 - \tau)$. Analogously, for the second argument of Equation 14 we get

$$(\tau + c\alpha) \log_2 d_{1,1} + (1 - \tau - c\alpha) \log_2 d_{0,1} = (\tau - \alpha) \log_2 d_{1,1} + (1 - \tau + \alpha) \log_2 d_{0,1} - (1+c)\alpha \underbrace{(\log_2 d_{0,1} - \log_2 d_{1,1})}_{\geq 0}$$

$$\geq -11 C_{\tau,\alpha},$$

where the inequality follows by an analogous computation. Therefore, the change of the weights when $j_t \neq a$ is in expectation at least $-11 C_{\tau,\alpha}$ when $c \in [0,1]$ and $\alpha \leq \frac{1}{2}\min(\tau, 1 - \tau)$. Let's consider now the case where $j_t = a$, the expected change is

$$p^t \log_2(d_{1,0}k + d_{1,1}(1 - k)) + (1 - p^t) \log_2(d_{0,0}k + d_{0,1}(1 - k)), \tag{15}$$

for some $k \in [0, 1]$. We have two cases: $k \leq q$ or $k > q$. When $k \leq q$ the coin flipped is $a$ then $p^1 \leq \tau$ and so $p^t \leq \tau + c\alpha$, in (Gretta & Price, 2024) it was shown that in this case Equation 15 is decreasing in $p^t$, then the minimum is

$$(\tau + c\alpha) \log_2(d_{1,0}k + d_{1,1}(1 - k)) + (1 - \tau - c\alpha) \log_2(d_{0,0}k + d_{0,1}(1 - k)) \qquad \text{if } k \leq q. \tag{16}$$

Conversely, when $k > q$ the coin flipped is $a + 1$ and then $p^1 \geq \tau$ so $p^t \geq \tau - c\alpha$. In this case the expression (15) is increasing in $p^t$ so the minimum is

$$(\tau - c\alpha) \log_2(d_{1,0}k + d_{1,1}(1 - k)) + (1 - \tau + c\alpha) \log_2(d_{0,0}k + d_{0,1}(1 - k)) \qquad \text{if } k > q. \tag{17}$$

In (Gretta & Price, 2024) the authors demonstrated that the minimum are obtained when $k \in \{0, 1\}$. Therefore, for $k = 1 > q$ we have Equation 17 while for $k = 0 < q$ we have instead Equation 16, which means that the minimum is

$$\min\left\{(\tau - c\alpha) \log_2 d_{1,0} + (1 - \tau + c\alpha) \log_2 d_{0,0} \, ; \, (\tau + c\alpha) \log_2 d_{1,1} + (1 - \tau - c\alpha) \log_2 d_{0,1}\right\},$$

which is at least $-11C_{\tau,\alpha}$ as demonstrated for the case $j_t \neq a$. To conclude, the expected change in Equation 12 is at least $12C_{\tau,\alpha}(1-\gamma)$, then the overall expected change for the potential is at least $12C_{\tau,\alpha}(1-\gamma) - 11C_{\tau,\alpha} = (1-12\gamma)C_{\tau,\alpha}$, cocnluding the proof. $\qquad\square$

The previous Lemma is the building block for the analysis of `BayesLearn`, as it allows the construction of a submartingale $\{Y_t\}_{t=1,\ldots}$ with $Y_{t+1} = \Phi_{t+1} - gt$, for $g = (1-12\gamma)C_{\tau,\alpha}$, that can be used to bound the probability to have a $\gamma$ fraction of good intervals, hence a positive potential. The analysis then follows directly from (Gretta & Price, 2024) with the distinction that the algorithm now with high probability in $B$ returns a $(\tau, \alpha(1+c))$-good coin, so proving Theorem E.2. Since the proof is identical (see Lemma 6 and Theorem 1 of (Gretta & Price, 2024)), we omit it. However, in order to make this paper self-contained, we will show a simple proof of Theorem 3.2 (which is much less general than Theorem E.2). We restate the theorem here.

**Theorem E.4.** *Let* $0 < \alpha \leq \frac{1}{4}$ *and suppose* $c \leq 1$ *There exists an algorithm for* `AdvMonotonicNBS`$(1/2, \alpha, c)$ *which uses* $O\left(\frac{\log B}{\alpha^2}\right)$ *coin flips and returns an* $(1/2, \alpha(1+c))$-*good with high probability in* $B$.

*Proof.* Let $\Phi$ be the potential function in Lemma E.3 in the case $\tau = 1/2$. Given Lemma E.3, by choosing $g = (1-12\gamma)C_{1/2,\alpha}$ equal to the lower bound of the lemma, we have that $\{Y_t\}_{t=1,\ldots}$, for $Y_{t+1} = \Phi_{t+1} - gt$, is a submartingale as

$$\mathbb{E}[Y_{t+1}|y_1,\ldots,y_t] = \mathbb{E}[\Phi_{t+1}|y_1,\ldots,y_t] - gt = \underbrace{\mathbb{E}[\Phi_{t+1} - \Phi_t|y_1,\ldots,y_t]}_{\geq g} - g + Y_t \geq Y_t$$

The difference of the martingale sequence $|Y_{t+1} - Y_t|$ is

$$|Y_{t+1} - Y_t| \leq |\log_2 w_{t+1}(a) - \log_2 w_t(a)| + 12C_{1/2,\alpha} + g \leq |\log_2 w_{t+1}(a) - \log_2 w_t(a)| + O(\alpha^2),$$

by triangle inequality and $C_{1/2,\alpha} = \Theta(\alpha^2)$ for $\alpha \in (0, 1/4)$ due to Lemma 10 (Gretta & Price, 2024). The remaining term is $|\log w_{t+1}(a) - \log w_t(a)| \leq \max\{\log d_{1,0}, \log d_{0,1}\} \leq O(\alpha)$ for Lemma 13 (Gretta & Price, 2024), thus $|Y_{t+1} - Y_t| \leq O(\alpha)$. We can use Azuma's inequality to bound the probability of having a negative potential

$$\begin{aligned}\Pr[\Phi_{t+1} \leq 0] &= \Pr[\Phi_{t+1} - gt - \Phi_1 \leq -gt - \Phi_1] \\ &= \Pr[Y_{t+1} - Y_0 \leq -gt - \Phi_1] \\ &\leq \exp\left(-\frac{(gt + \Phi_1)^2}{t \cdot O(\alpha^2)}\right) \quad \text{for } gt \geq -\Phi_1.\end{aligned}$$

Note that $\Phi_1 = -\log(B-1)$. Therefore, picking $T = O\left(\frac{\log B}{g}\right)$ sufficiently large, we get that $\frac{(gT+\Phi_1)^2}{T \cdot O(\alpha^2)} \geq \lambda \log B$ for any desired constant $\lambda > 0$. Thus,

$$\Pr[\Phi_{T+1} \leq 0] \leq B^{-\lambda}.$$

On the other hand, note that if $\Phi_{T+1} > 0$, then

$$0 < \frac{\Phi_{T+1}}{12C_{1/2,\alpha}} \leq (|\{x \in L : x \in [\ell, r]\}| - \gamma|L|),$$

and so, a $\gamma$ fraction of the intervals in $L$ are $(1/2, \alpha(1+c))$-good. Now we can order the intervals in $L$ in sorted order according to their indices $i$ of the corresponding coins. By picking a subset $S$ of every $(1/\gamma)$th of them, we are ensured that one of them will be good (conditional on the high probability event $\Phi_{T+1} > 0$). For each interval in $S$, we can test whether it is $(1/2, \alpha(1+c))$-good with high probability using $O(\frac{\log B}{\alpha^2})$ coin flips of each of the coins at its endpoints. Therefore, we successfully determine an $(1/2, \alpha(1+c))$-good coin with high probability in $B$. If we pick $\gamma = 1/13$, the total number of coins flipped is

$$T + |S|O\left(\frac{\log B}{\alpha^2}\right) = O\left(\frac{\log B}{g}\right) + O\left(\frac{\log B}{\alpha^2}\right) = O\left(\frac{\log B}{\alpha^2}\right),$$

where the final bound uses that $g = (1-12\gamma)C_{1/2,\alpha} = \frac{1}{13}C_{1/2,\alpha} = \Theta(\alpha^2)$. This completes the proof. $\qquad\square$

# F. Lower Bounds

In this section, we give the proof of our lower bounds: Theorem 1.2 for adaptive mechanisms, and Theorem 1.3 for non-adaptive mechanisms. Since an algorithm for the median problem implies an algorithm for a general quantile by inserting dummy elements, we will correspondingly show a lower bound for the general quantile problem, demonstrating that all quantiles (not too close to the minimum or maximum) are as hard as the median. We use the notation `LDPstat-quantile` and `LDPemp-quantile` with the additional parameter $q$ to describe the corresponding generalizations.

## F.1. Adaptive Mechanisms

In the statistical setting, our building block will be the lower bound framework in (Duchi et al., 2013), which turns the estimation problem into a distinguishing problems. In our setting, it means that if a mechanism attains low error on the quantile problem, then it is good at distinguishing distributions with different $q$th quantiles from each other, even from a "hard" family of distributions. Our hard family of distributions will be close in statistical distance, but still have different $q$th quantiles:

$$P_\beta(i) = \begin{cases} q - 2\alpha & i = 1 \\ 4\alpha & i = \beta \\ 1 - q - 2\alpha & i = B, \end{cases}$$

for $\beta \in \{2, \ldots, B-2\}$. If $\beta$ is chosen uniformly at random, then our LDP distinguishing mechanism will be able to deliver $\log(B)$ bits of information (measured with the mutual information), by Fano's inequality. However, there is an upper bound on the amount of mutual information possible with an LDP protocol, as first established in (Duchi et al., 2013). This gives us the following bound:

**Theorem F.1.** *Let $B \geq 4$, $\alpha < \frac{1}{2}$, $\varepsilon < 1$, and $q \in (2\alpha, 1-2\alpha)$. Suppose there is a sequentially interactive $\varepsilon$-LDP algorithm that for any distribution $\mathcal{D}$ solves* `LDPstat-quantile`$(\mathcal{D}, n, \alpha, \varepsilon, q)$ *with probability at least $\frac{1}{2}$. Then*

$$n \geq \Omega\left(\frac{\log B}{\varepsilon^2 \alpha^2}\right).$$

*Proof.* Let $\mathcal{M} = (\mathcal{M}_1, \ldots, \mathcal{M}_n)$ be a sequentially interactive $\varepsilon$-LDP protocol which solves `LDPstat-quantile`$(\mathcal{D}, n, \alpha, \varepsilon, q)$ with probability $\geq 1/2$, i.e., for a distribution $\mathcal{D}$ and $n$ samples $x_1, \ldots, x_n$ from $\mathcal{D}$, it outputs an estimate $\tilde{m} = \mathcal{M}(x_1, \ldots, x_n)$ which is an $\alpha$-approximation to the true median of $\mathcal{D}$ with probability at least $1/2$. Consider the following collection of distributions $\{P_\beta\}_{\beta \in [B]}$ indexed by a parameter $\beta \in \{1, \ldots, B-1\}$, with probability mass functions defined by

$$P_\beta(i) = \begin{cases} q - 2\alpha & i = 0 \\ 4\alpha & i = \beta \\ 1 - q - 2\alpha & i = B - 1. \end{cases}$$

Let $\beta^*$ be uniformly at random from $\{1, \ldots, B-1\}$, and generate $n$ samples $x_1, \ldots, x_n$ from $P_{\beta^*}$. Let $y_i = \mathcal{M}_i(x_i, y_1, \ldots, y_{i-1})$ be the $\varepsilon$-differentially private output of user $i$ generated in the manner of in the manner of (2). Finally, let $\tilde{m}_{\beta^*} = \mathcal{F}(y_1, \ldots, y_n)$ be the estimated median output by our protocol. By Fano's inequality,

$$I(\beta^*; y_1, \ldots, y_n) \geq H(\beta^*) - H(\mathbf{1}[\tilde{m}_{\beta^*} = \beta^*]) +$$
$$- \Pr[\tilde{m}_{\beta^*} \neq \beta^*] \log_2(B-1)$$
$$\geq \log_2(B) - 1 - \frac{1}{2}\log(B-1)$$
$$\geq \frac{1}{4}\log_2(B),$$

where $I$ denotes the mutual information and $H$ denotes the binary entropy.

However, using the fact that our mechanism is $\varepsilon$-LDP, we can use the *upper bound* from (Duchi et al., 2013) on the mutual information between $\{y_1, \ldots, y_n\}$ and $\beta^*$. According to their bound (see the calculations following Corollary 1 of (Duchi

et al., 2013)) for all $\varepsilon < 1$, we have

$$I(\beta^*; y_1, \ldots, y_n) \leq 4(e^\varepsilon - 1)^2 \frac{n}{B^2} \sum_{\beta, \beta' \in [B]} \|P_\beta - P_{\beta'}\|_{TV}^2.$$

Note that the total variation distance between $P_\beta$ and $P_{\beta'}$ for $P_\beta \neq \beta'$ is $4\alpha$. It follows that,

$$I(\beta^*; y_1, \ldots, y_n) \leq 256\varepsilon^2 n\alpha^2.$$

Combining inequalities, we see that $\log(B) \leq 1024\varepsilon^2\alpha^2 n$ which implies that $n = \Omega(\frac{\log B}{\varepsilon^2\alpha^2})$, as desired. $\qquad\square$

To adapt this to the empirical setting, observe that any algorithm for the empirical problem can be used to solve the statistical problem by sampling, and then applying the empirical algorithm.

**Theorem F.2.** *Let $B \geq 4, \alpha < \frac{1}{2}$, and $q \in (2\alpha, 1 - 2\alpha)$. If $\varepsilon \leq \min\{1, \frac{1}{64}\sqrt{\log B}\}$, then any sequentially interactive $\varepsilon$-LDP algorithm that solves* `LDPemp-quantile`$(\{x_i\}_{i=1}^n, \alpha, \varepsilon, q)$ *with probability $\frac{3}{4}$ requires $n \geq \Omega(\frac{\log B}{\varepsilon^2\alpha^2})$.*

*Proof.* We will first show any algorithm $\mathcal{M}$ which solves `LDPemp-quantile`$(\{x_i\}_{i=1}^n, \alpha, \varepsilon, q)$ with probability $\frac{3}{4}$ can be used to solve `LDPstat-quantile`$(\mathcal{D}, n, 2\alpha, \varepsilon, q)$ with probability $\frac{1}{2}$. The algorithm will simply apply $\mathcal{M}$ to the sampled dataset $\{x_1, \ldots, x_n\}_{i=1}^n$ from $\mathcal{D}$. Using Hoeffding's bound, together with the fact that $n \geq \frac{2}{\alpha^2}$, the $q$th quantile $x_{(q)}$ of the sampled dataset will have quantile error at most $\alpha$ from the true $q$th quantile of $\mathcal{D}$ with probability at least $\frac{3}{4}$. Thus, the correctness guarantee of $\mathcal{M}$ carries over, with success probability at least $\frac{1}{2}$ by the union bound.

Using the above reduction, we are able to show a lower bound of $n \geq \Omega(\frac{\log B}{\varepsilon^2\alpha^2})$ so long as these $\frac{\log(B)}{1024\varepsilon^2\alpha^2} \geq \frac{4}{\alpha^2}$ is satisfied. $\qquad\square$

Theorem 1.2 follows directly from Theorem F.2. These lower bounds establish that our algorithm in Theorem 1.1 is tight in the $\varepsilon = O(1)$ regime.

### F.2. Non-Adaptive Mechanisms

In order to prove non-adaptive lower bound of Theorem 1.3, we will apply a lower bound for learning a cumulative distribution function from (Edmonds et al., 2020). The CDF learning problem is defined as follows:

**Definition F.3.** *Given a dataset $X = \{x_1, \ldots, x_n\} \subseteq [B]$, let $F_X(t)$ denote its c.d.f. (given by $F_X(t) = \frac{1}{n}\sum_{i=1}^n \mathbf{1}[x_i \leq \tau]$. In the* `LDPemp-cdf`$(\{x_i\}_{i=1}^n, \alpha, \varepsilon)$ *problem, the task is to output a function $\tilde{F}$ under $\varepsilon$-LDP which approximates the c.d.f. up to error alpha at all points; i.e.*

$$\mathbb{E}[\|\tilde{F} - F_X\|_\infty] \leq \alpha.$$

Observe the above definition considers the expected error, which different from applying Definitions 2.3 or 2.2 with constant probability of failure. We change to expectation because it is the setting considered by (Edmonds et al., 2020), but may easily convert between the two types of guarantees (since the maximum c.d.f. error is 1, we only need failure probability of $\alpha$ to obtain a bound on the expectation).

A lower bound on $n$ for learning a c.d.f. was shown in (Edmonds et al., 2020) for $\varepsilon < 1$:

**Theorem F.4.** *(Theorem 23 in (Edmonds et al., 2020)) There exists a constant $C$ such that, for all $\alpha$ sufficiently small, and all $\varepsilon < 1$ and $B$ satisfying $\frac{\log^2(B)}{\varepsilon^2\alpha^2\log(1/\alpha)^2} \geq \frac{C\log(2B)}{\alpha^2} + \frac{C}{\varepsilon^2\alpha^2}$ any $\varepsilon$-LDP algorithm which solves* `LDPemp-cdf`$(\{x_i\}_{i=1}^n, \alpha, \varepsilon)$ *requires*

$$n \geq \Omega\left(\frac{\log^2(B)}{\varepsilon^2\alpha^2\log(1/\alpha)^2}\right).$$

In particular, it is sufficient to satisfy Theorem F.4 when $\alpha \geq B^{-\Omega(1)}$ and $\varepsilon \leq \frac{\sqrt{\log B}}{\log(1/\alpha)}$, which is a mild assumptions as $\alpha, \varepsilon$ are constants typically significantly less than $B$.

To apply this theorem, we prove a reduction from `LDPemp-quantile` for constant $q$ to `LDPemp-cdf`. The c.d.f. may be solved to accuracy $\alpha$ by computing the $\alpha, 2\alpha, \ldots, 1 - \alpha$ quantiles. With correct padding, we may use our

`LDPemp-quantile` algorithm to answer any of these quantiles. To answer all $\frac{1}{\alpha}$ quantiles, our reduction uses non-adaptivity in a crucial way: it is only necessary to collect responses once, add in the proper amount of responses for padded elements, and then post-process them into a response. By boosting the accuracy of the quantile with $\log(\frac{1}{\alpha})$ runs, each quantile estimate will have success probability at least $1 - \alpha^2$. We may then apply a union bound to obtain a bound on the expected error on all quantiles (giving the desired c.d.f. error).

**Lemma F.5.** *Suppose there is a non-adaptive algorithm solving* `LDPemp-quantile`$(\{x_i\}_{i=1}^n, \alpha, \varepsilon, q)$ *with probability* $\frac{3}{4}$, *for all datasets of size* $n \geq n_0$. *Then, there is a non-adaptive algorithm solving* `LDPemp-cdf`$(\{x_i\}_{i=1}^n, \alpha(1+\frac{1}{q}), 2\varepsilon \log(\frac{1}{\alpha}))$ *with probability* $\frac{3}{4}$ *for any datasets of size* $n \geq n_0$ .

*Proof.* WLOG, we may assume that $q \leq \frac{1}{2}$. Suppose we are given a dataset $\{x_1\}_{i=1}^n$. We will first show how to estimate any quantile $q'$ with success probability at least $\frac{3}{4}$ and error $\frac{\alpha}{q}$. We may do this by adding padded elements to the dataset. Specifically, if $q < q'$, then adding $\frac{q'-q}{q}n$ padded $B$s to the dataset will ensure that the $q$th quantile will match the $q'$th quantile of the original dataset. If $q' < q$, then adding $\frac{q-q'}{1-q}n$ padded 1s to the dataset will ensure the $q$th quantile will match the $q'$th quantile of the original dataset. Observe that quantile error of $\alpha$ in the padded dataset corresponds to quantile error $\frac{\alpha}{q}$ in the original dataset.

Next, observe that computing the quantiles $\{\alpha, 2\alpha, \ldots, 1-\alpha\}$ with error $\alpha$ gives a $2\alpha$ c.d.f. estimation. We will simulate running the above procedure on all $\frac{1}{\alpha}$ quantiles by collecting the responses $r_1 = \mathcal{M}_1(x_1), \ldots, r_n = \mathcal{M}_n(x_n)$, and then $r_i^0 = \mathcal{M}_i(1), r_i^1 = \mathcal{M}_i(B)$ for $n+1 \leq i \leq n + \frac{n}{q}$. By picking the correct padded elements, we may post-process the response to obtain an estimate of any $q'$th quantile with error at most $\frac{\alpha}{q}$. This crucially uses the fact that the $\mathcal{M}$ are non-interactive, as any state shared between the $\mathcal{M}_i$ could not be simulated for all runs at once.

Each of the above estimated quantiles has error $\frac{\alpha}{q}$ with probability at least $\frac{3}{4}$. We may boost the success probability by running the above procedure independently $2\log(\frac{1}{\alpha})$ times and taking the *median* quantile estimate; by Chernoff's bound, we are guaranteed that each quantile estimate is at most $\frac{\alpha}{q}$ from the true $q'$ quantile with probability $1 - \alpha^2$. By the union bound all quantile estimates will have error $\frac{\alpha}{q}$ with success probability at least $1 - \alpha$, and the expected c.d.f. error is at most $\alpha + (1-\alpha)\frac{\alpha}{q} \leq \alpha(1+\frac{1}{q})$. Finally, the privacy parameter is $2\varepsilon \log(\frac{1}{\alpha})$ by simple composition. $\square$

Theorem F.4 and Lemma F.5 together give us a lower bound on the median error under LDP:

**Theorem F.6.** *For any constant* $q \in (0.1, 0.9)$, *any* $\alpha$ *sufficiently small,* $\varepsilon < \frac{1}{\log(1/\alpha)}$, *and* $B$ *such that* $\alpha \geq B^{-\Omega(1)}$ *and* $\varepsilon \leq \frac{\sqrt{\log B}}{\log(1/\alpha)^2}$, *any* $\varepsilon$-*LDP algorithm which solves* `LDPquantile-emp`$(\{x_i\}_{i=1}^n, \alpha, \varepsilon, q)$ *with probability at least* $\frac{3}{4}$ *requires*

$$n \geq \Omega\left(\frac{\log^2(B)}{\varepsilon^2 \alpha^2 \log(1/\alpha)^4}\right).$$

Theorem 1.3 follows directly from this result.

## G. Naive Shuffle-DP binary Search for the Median

This section is dedicated to proving Theorem 1.4.

The naive binary search with errors algorithm Algorithm 3 tests each coin up to $\alpha$-accuracy and a $\beta/\log B$ failure probability, such that a simple union bound over all $\log B$ steps of binary searching will yield an $(\alpha, \beta)$-accurate estimate. This algorithm is suboptimal up to logarithmic factors, although there are indications that its strong constant factors can make up the difference in some parameter regimes (Karp & Kleinberg, 2007; Gretta & Price, 2024). The simple fact that this algorithm runs in deterministic number of rounds, with a deterministic number of samples per round, allows for a straightforward application of amplification by shuffling (Feldman et al., 2021), something we could not achieve with the fully adaptive Bayesian updates algorithm.

Algorithm 3 provides the functional details of this algorithm, which we analyze below to find the necessary sample complexity $n$ for a desired accuracy $\alpha$ and failure probability $\beta$. The shuffler is not included in the pseudocode, as it has no effect on the accuracy of the algorithm. To compute the final sample complexity, it will be most important to prove how accurately we can learn the bias of one coin at each step of the algorithm. We give a tight characterisation in Lemmas G.1 and G.2

We consider both statistical error, where samples are assumed to be drawn from some unknown distribution with mean $p$, and we are interested in an estimate $\hat{p}$ which is close to that true mean, and the empirical setting where we make no assumption on the distribution of the samples, and are interested in how close our estimate $\hat{p}$ is to the "best-case" sample mean $\frac{1}{n}\sum_{i=1}^{n} x_i$.

**Lemma G.1** (Sample complexity of learning one coin to its statistical mean.)**.** *Given samples $\{x_i\}_{i=1}^{n}$ from a Bernoulli random variable $X$ with mean $p$ received through a binary randomized response channel $\mathcal{M}$ such that $y_i \sim \mathcal{M}(x_i)$, we can estimate $\hat{p} = \frac{1}{n}\frac{e^\varepsilon+1}{e^\varepsilon-1}\sum_{i=1}^{n} y_i - \frac{1}{e^\varepsilon-1}$. In order to learn an $(\alpha, \beta)$-estimate of $p$, $\Pr[|\hat{p}-p| > \alpha] < \beta$ it suffices to use $n$ samples where,*

$$n \le \left(\frac{2p(1-p)}{\alpha^2} + \frac{e^\varepsilon}{\alpha^2(e^\varepsilon-1)^2} + \frac{2(e^\varepsilon+1)}{4\alpha(e^\varepsilon-1)}\right)\log(1/\beta).$$

*In other words, the sample complexity of learning one coin to its statistical mean with constant failure probability is $O\left(\frac{1}{\alpha^2\varepsilon^2} + \frac{p(1-p)}{\alpha^2}\right)$, when $\varepsilon < 1$, or $O\left(\frac{1}{\alpha^2 e^\varepsilon} + \frac{p(1-p)}{\alpha^2}\right)$, when $\varepsilon \ge 1$.*

*Proof.* Given a Bernoulli random variable $x$ with mean $p$, and a binary randomized response channel $\mathcal{M}$ (see Lemma A.1) the distribution induced by applying $\mathcal{M}$ to $x$ is:

$$y = \mathcal{M}(x) \sim \text{Bern}\left(\frac{e^\varepsilon}{e^\varepsilon+1}p + (1-p)\frac{1}{e^\varepsilon+1}\right) = \text{Bern}\left(\frac{e^\varepsilon-1}{e^\varepsilon+1}p + \frac{1}{e^\varepsilon+1}\right).$$

The variance of this distribution is

$$\sigma^2 = \text{Var}(y) = \left(\frac{1}{e^\varepsilon+1} + \frac{e^\varepsilon-1}{e^\varepsilon+1}p\right)\left(\frac{e^\varepsilon}{e^\varepsilon+1} - \frac{e^\varepsilon-1}{e^\varepsilon+1}p\right)$$

$$= \left(\frac{e^\varepsilon-1}{e^\varepsilon+1}\right)^2 p(1-p) + \frac{e^\varepsilon}{(e^\varepsilon+1)^2}.$$

We then proceed by simple rearranging, substitution, and application of Bernstein's inequality Theorem A.3.

$$\Pr\left[|\hat{p}-p| > \alpha\right] = \Pr\left[\left|\frac{1}{n}\frac{e^\varepsilon+1}{e^\varepsilon-1}\sum_{i=1}^{n} y_j - \frac{1}{e^\varepsilon-1} - \left(\frac{e^\varepsilon+1}{e^\varepsilon-1}\mathbb{E}\left[y\right] - \frac{1}{e^\varepsilon-1}\right)\right| > \alpha\right]$$

$$= \Pr\left[\left|\frac{e^\varepsilon+1}{e^\varepsilon-1}\left(\frac{1}{n}\sum_{i=1}^{n} y_i - \mathbb{E}\left[y\right]\right)\right| > \alpha\right]$$

$$= \Pr\left[\left|\frac{1}{n}\sum_{j=1}^{n} y - \mathbb{E}\left[y\right]\right| > t\right] \hspace{3cm} \left(t = \alpha\frac{e^\varepsilon-1}{e^\varepsilon+1}\right)$$

$$\beta \le \exp\left(\frac{-nt^2}{2\sigma^2 + \frac{2t}{3}}\right) \hspace{4cm} \text{(Bernstein's Inequality)}$$

$$n \le \left(\frac{2\sigma^2}{t^2} + \frac{2}{3t}\right)\log(1/\beta)$$

$$= \left(\frac{2p(1-p)}{\alpha^2} + \frac{2e^\varepsilon}{\alpha^2(e^\varepsilon-1)^2} + \frac{2(e^\varepsilon+1)}{3\alpha(e^\varepsilon-1)}\right)\log(1/\beta). \hspace{1cm} \text{(Substituting } t \text{ and } \sigma^2\text{)}$$

$\square$

**Lemma G.2** (Sample complexity of learning one coin to its sample mean.)**.** *Given samples $\{x_i\}_{i=1}^{n}$ where each $x_i \in \{0,1\}$, and private outputs $y_i \sim \mathcal{M}(x_i)$, the true sample mean is $P = \frac{1}{n}\sum_{i=1}^{n} x_i$. Denote the sample mean of the collected private outputs $Y = \frac{1}{n}\sum_{i=1}^{n} y_i$. Our estimator of the sample mean will be similar to the statistical case, where $\widehat{P} = \frac{e^\varepsilon+1}{e^\varepsilon-1}Y - \frac{1}{e^\varepsilon-1}$. In order to learn an $(\alpha, \beta)$-estimate of $P$ it is sufficient to use $n$ samples such that*

$$n \le \left(\frac{2e^\varepsilon}{\alpha^2(e^\varepsilon-1)^2} + \frac{2(e^\varepsilon+1)}{3\alpha(e^\varepsilon-1)}\right)\log(1/\beta).$$

*Therefore, the sample complexity of learning the sample mean with constant failure probability is $O\left(\frac{1}{\alpha^2 \varepsilon^2}\right)$, when $\varepsilon < 1$, or $O\left(\frac{1}{\alpha^2 e^\varepsilon}\right)$, when $\varepsilon \geq 1$. It is pleasing to note that this recovers the sample complexity of learning in the statistical case, up to the additive sampling error.*

*Proof.* The proof will proceed similarly to the statistical case. The key difference will be the variance of $y$ in this case which is

$$\sigma^2 = \sigma^2(\mathcal{M}(0)) = \sigma^2(\mathcal{M}(1)) = \frac{e^\varepsilon}{(e^\varepsilon + 1)^2}.$$

The derivation then proceeds as in the statistical case.

$$
\begin{aligned}
\Pr[|\widehat{P} - P| > \alpha] &= \Pr\left[\left|\frac{e^\varepsilon + 1}{e^\varepsilon - 1} Y - \frac{1}{e^\varepsilon - 1} - P\right| > \alpha\right] \\
&= \Pr\left[\left|\frac{e^\varepsilon + 1}{e^\varepsilon - 1} Y - \frac{1}{e^\varepsilon - 1} - \left(\frac{e^\varepsilon + 1}{e^\varepsilon - 1}\mathbb{E}\left[Y\right] - \frac{1}{e^\varepsilon - 1}\right)\right| > \alpha\right] \\
&= \Pr\left[\left|\frac{e^\varepsilon + 1}{e^\varepsilon - 1}\right| \left(Y - \mathbb{E}\left[Y\right]\right) > \alpha\right] \\
&= \Pr\left[\left|Y - \mathbb{E}\left[Y\right]\right| > t\right] && \left(t = \alpha\frac{e^\varepsilon - 1}{e^\varepsilon + 1}\right) \\
&\leq \exp\left(\frac{-nt^2}{2\sigma^2 + \frac{2t}{3}}\right) && \text{(Bernstein's Inequality)} \\
n &\leq \left(\frac{2\sigma^2}{t^2} + \frac{2}{3t}\right)\log(1/\beta) \\
&= \left(\frac{2e^\varepsilon}{\alpha^2(e^\varepsilon - 1)^2} + \frac{2(e^\varepsilon + 1)}{3\alpha(e^\varepsilon - 1)}\right)\log(1/\beta). && \text{(Substituting } t \text{ and } \sigma^2)
\end{aligned}
$$

$\square$

With this we can now formally state the sample complexity of a naive binary search for the median under local differential privacy. We will focus on the empirical case for this result.

**Theorem G.3** (Naive Binary Search for the Median under Local Differential Privacy). *The naive algorithm as described by Karp & Kleinberg (2007), under the constraints of $\varepsilon$-local differential privacy, has sample complexity*

$$n \leq \left(\frac{2e^\varepsilon}{\alpha^2(e^\varepsilon - 1)^2} + \frac{2(e^\varepsilon + 1)}{3\alpha(e^\varepsilon - 1)}\right)\log(B)\log\left(\frac{\log B}{\beta}\right).$$

*We can therefore say that for $\varepsilon < 1$, the naive approach has sample complexity $O\left(\frac{\log B}{\alpha^2 \varepsilon^2}\log\left(\frac{\log B}{\beta}\right)\right)$, and for $\varepsilon \geq 1$ it has sample complexity $O\left(\frac{\log B}{\alpha^2 e^\varepsilon}\log\left(\frac{\log B}{\beta}\right)\right)$.*

*Proof.* Given $n$ total users, let $n' = n/\log(B)$ and let $\beta' = \beta/\log(B)$, apply Lemma G.2 with $n', \beta'$ to get sample complexity.

$$n \leq \left(\frac{2e^\varepsilon}{\alpha^2(e^\varepsilon - 1)^2} + \frac{2(e^\varepsilon + 1)}{3\alpha(e^\varepsilon - 1)}\right)\log(B)\log\left(\frac{\log B}{\beta}\right).$$

By a union bound over all $\log B$ rounds of the binary search, the final estimate will be an $(\alpha, \beta)$-approximate median. $\square$

As stated in the introduction, the primary motivation for this approach is that by dividing the algorithm into a few deterministic stages, with many samples tested at each stage, we can hope to apply amplification by shuffling (Feldman et al., 2021). We state the amplification by shuffling result here, and a subsequent lemma that will be useful to our analysis.

**Theorem G.4** (Feldman, McMillan, and Talwar (2021, Theorem 3.1)). *For any domain $\mathcal{X}$, let $\mathcal{M}_t : \mathcal{M}_1 \times \ldots \times \mathcal{M}_{t-1} \times \mathcal{X} \to \mathcal{Y}$ for $t \in [n]$ be a sequence of randomizers such that $\mathcal{M}_t(y_{1:t-1}, \cdot)$ is $\varepsilon_L$-local DP; and let $S$ be the algorithm that given*

*a tuple of $n$ messages, outputs a uniformly random permutation of said messages. Then for any $\delta \in (0, 1]$ such that $\varepsilon_L \leq \log \frac{n}{16 \log(2/\delta)}$, $S \circ \mathcal{Y}^n$ is is $(\varepsilon, \delta)$-DP, where*

$$\varepsilon \leq \log \left( 1 + 8 \frac{e^{\varepsilon_L} - 1}{e^{\varepsilon_L} + 1} \left( \sqrt{\frac{e^{\varepsilon_L} \log(4/\delta)}{n}} + \frac{e^{\varepsilon_L}}{n} \right) \right)$$

This implies the following useful lemma,

**Lemma G.5** (Amplification by shuffling). *Fix any $\delta \in (0, 1]$, $\varepsilon \in (0, 1]$, and $n$ such that $\varepsilon > 16\sqrt{\log(4/\delta)/n}$. Then, for*

$$\varepsilon_L := \log \frac{\varepsilon^2 n}{80 \log(4/\delta)}$$

*Shuffling the messages of $n$ users using the same $\varepsilon_L$-LDP randomizer satisfies $(\varepsilon, \delta)$-shuffle differential privacy.*

*Proof.* For $\varepsilon, \delta$ and $\varepsilon_L$ as above we have $0 < \varepsilon_L \leq \log \frac{n}{16 \log(2/\delta)}$. Applying Theorem G.4, we get $(\varepsilon', \delta)$-differential privacy for

$$\varepsilon' \leq \log \left( 1 + 8 \underbrace{\frac{e^{\varepsilon_L} - 1}{e^{\varepsilon_L} + 1}}_{<1} \underbrace{\left( \frac{\varepsilon}{\sqrt{80}} + \frac{\varepsilon^2}{80 \log(4/\delta)} \right)}_{<\varepsilon/8} \right) \leq \varepsilon$$

Proving the lemma. $\qquad\square$

We can now prove Theorem 1.4.

**Theorem G.6** (Restatement of Theorem 1.4). *Let $r = \log_2 B$. There exists a protocol for `shuffle-emp-`-`median`($\{x_i\}_{i=1}^n, \alpha, \varepsilon, \delta, r$) with success probability $1 - \beta$ provided that*

$$n = O\left( \left( \frac{1}{\alpha^2} + \frac{1}{\varepsilon^2} \right) \log B \sqrt{\log(1/\delta) \log \frac{\log B}{\beta}} \right).$$

*The protocol has $r = \log_2 B$ rounds of adaptivity and queries shuffled batches of $n/\log_2(B)$ users.*

*Proof of Theorem 1.4.* Take the sample complexity achieved in Theorem G.3, and note that we are in the $\varepsilon \gg 1$ regime as we will be applying taking $\varepsilon_L \in O(\log n)$. We therefore have

$$n = O\left( \frac{\log B}{\alpha^2 e^{\varepsilon_L}} \log \frac{\log B}{\beta} \right)$$

We apply Lemma G.5 while noting that at each stage we shuffle $n' = n/\log(B)$ users. Setting $\varepsilon_L = \log \frac{\varepsilon^2 n}{80 \log(B) \log(4/\delta)}$ and rearranging gives that for each step of the binary search we have enough users to accurately learn the CDF of the remaining suffix of users within error $\alpha/2$ with probability $\beta/\log B$. Union bounding over all $\log B$ steps of the binary search, we conclude that with probability $1 - \beta$, every step succeeds. This gives sample complexity,

$$n = O\left( \frac{\log B}{\alpha \varepsilon} \sqrt{\log(1/\delta) \log \frac{\log B}{\beta}} \right),$$

but we are not finished. We have to handle the multiple restrictions on parameter regimes

$$O\left( \frac{\log B}{\alpha \varepsilon} \sqrt{\log(1/\delta) \log \frac{\log B}{\beta}} \right) \geq n > \max\left\{ \frac{\log B}{\alpha^2}, \frac{\log B \log(1/\delta)}{\varepsilon^2} \right\}.$$

---

**Algorithm 3** Shuffled Noisy Binary Search for quantile estimation.

---

1: **function** $\text{RR}_\varepsilon(X)$
2:    **return** $X$ w.p. $\frac{e^\varepsilon}{e^\varepsilon+1}$ **otherwise** $\neg X$
3: **end function**

4: **function** Shuffled Noisy Binary Search$(\{x_i\}_{i=1}^n, B, \tau, \varepsilon)$
5:    $X \leftarrow \pi(\{x_i\}_{i=1}^n)$ {Random permutation of the users}
6:    $\varepsilon' \leftarrow \log \frac{\varepsilon^2 n}{80 \log(4/\delta)}$
7:    $r \leftarrow \lceil \log_2 B \rceil$ {Number of rounds}
8:    $s \leftarrow \lfloor n/r \rfloor$ {Batch size}
9:    $L \leftarrow 1$
10:   $R \leftarrow B$
11:   **for** $j = 1, 2, \ldots, r$ **do**
12:      $m \leftarrow L + \lfloor (R - L)/2 \rfloor$
13:      $S_j \leftarrow X_{l=(j-1)s+1}^{js}$ {Sample one batch}
14:      $\xi \leftarrow 0$
15:      **for** $k = 1, 2, \ldots, s$ **do**
16:         $y_k \leftarrow [S_{j,k} \leq m]$ {Query}
17:         $\tilde{y}_k \leftarrow \text{RR}_{\varepsilon'}(y_i)$ {Apply Randomized Response with $\varepsilon'$}
18:         $\xi \leftarrow \xi + \tilde{y}_k$
19:      **end for**
20:      $\hat{p} \leftarrow \frac{1}{s} \frac{e^{\varepsilon'}+1}{e^{\varepsilon'}-1}\xi - \frac{1}{e^{\varepsilon'}-1}$
21:      **if** $\hat{p} < \tau$ **then**
22:         $L \leftarrow m + 1$
23:      **else**
24:         $R \leftarrow m - 1$
25:      **end if**
26:   **end for**
27:   **return** $m$
28: **end function**

---

The right hand side of this inequality comes from restrictions present in Lemmas 3.1 and G.5 on $n$ and $\varepsilon$ respectively, the latter comes from using $n' = n/\log(B)$ in the restriction on $\varepsilon$. A trivial solution is be to take $1/(\alpha\varepsilon)$ and replace it with $1/\min\{\alpha^2, \varepsilon^2\}$, which gives

$$n = O\left(\left(\frac{1}{\alpha^2} + \frac{1}{\varepsilon^2}\right) \log B \sqrt{\log(1/\delta) \log \frac{\log B}{\beta}}\right).$$

$\square$

This result has an improved dependence in $\varepsilon$ and $\alpha$, and could be preferable from a communication perspective. Rounds of adaptivity are a restricting factor in distributed learning, and our goal was to understand the trade offs possible under privacy constraints. It is of practical interest to know whether the constraint on $n$ in Lemma G.5 can be improved from $n = \Omega\left(\log(1/\delta)/\varepsilon^2\right)$ to $\Omega\left(\log(1/\delta)/\varepsilon\right)$. This, in combination with a strengthening of Lemma 3.1 to have a linear dependence on $\alpha$, would allow the analysis to go through with only a $1/(\alpha\varepsilon)$ dependence.

## H. A Note on the Continuous Case

If we replace the discrete domain $[B]$ with a continuous one, say $[0, 1]$, it is generally impossible to obtain quantile error $o(1)$ using a finite number of samples under LDP. This follows from our lower bounds by discretizing $[0, 1]$ into $[B]$ buckets and letting $B \to \infty$. In fact, this is a general issue for quantile or range estimation problems in DP (even beyond the local model), which is why related work studies the discrete setting (Beimel et al., 2016; Bun et al., 2015; Kaplan et al., 2020; Cormode et al., 2019). On a more positive note, if we impose mild guarantees on the family of possible distributions the

---

**Algorithm 4** `DPBayeSS` for empirical quantile estimation with local differential privacy, from Algorithm 3 in (Gretta & Price, 2024)

---

1: **function** ReductionToGamma($\{x_i\}_{i,\ldots,M}, B, \alpha, \varepsilon, \gamma$)
2:    $L \leftarrow$ DPBayesLearn($\{x_i\}_{i,\ldots,M}, B, \frac{1}{2}, \alpha, M, \varepsilon$) {From Algorithm 2 with $\varepsilon$-RR on each coin flip}
3:    $R \leftarrow \{\}$ {Get the $\gamma$-quantiles of $L$}
4:    **for** $i \in \left[\left\lfloor \frac{|L|}{\lceil \gamma |L| \rceil} \right\rfloor \right]$ **do**
5:       append $L_{\lceil \gamma |L| \rceil i}$ to $R$
6:    **end for**
7:    **return** Sorted(RemoveDuplicates($R$))
8: **end function**

9: **function** DPBayeSS($\{x\}_{i,\ldots,n}, B, \varepsilon$)
10:    $M_{B_1} \leftarrow \left\lfloor \frac{n \log(B)}{\log(B) + \log\log(B) + 1} \right\rfloor$
11:    $M_{B_2} \leftarrow \left\lfloor \frac{n \log\log(B)}{\log(B) + \log\log(B) + 1} \right\rfloor$
12:    $\tilde{\alpha} \leftarrow 0.6\sqrt{\frac{\log B}{n}}$ {Hyper-parameter obtained empirically in Section I.1}
13:    $R \leftarrow$ ReductionToGamma($\{x_i\}_{i=1,\ldots,M_{B_1}}, B, \tilde{\alpha}, \varepsilon, \frac{1}{\log^2 B}$)
14:    **if** $|R| > 13$ **then**
15:       $R \leftarrow [1] + R + [n]$ {Pad $R$ with the extremes of the initial problem.}
16:       $R \leftarrow$ ReductionToGamma($\{x_i\}_{i=M_{B_1}+1,\ldots,M_{B_1}+M_{B_2}}, R, \tilde{\alpha}, \varepsilon, \frac{1}{13}$) {Reducing $R$ to fixed size $|R| \leq 13$}
17:       **return** Apply NoisyBinarySearch with $\varepsilon$-RR over the coins in $R$ using dataset $\{x_i\}_{i=M_{B_1}+M_{B_2}+1,\ldots,n}$ to find probability closest to $\frac{1}{2}$
18:    **else**
19:       **return** Apply NoisyBinarySearch with $\varepsilon$-RR over the coins in $R$ using dataset $\{x_i\}_{i=M_{B_1}+1,\ldots,n}$ to find probability closest to $\frac{1}{2}$
20:    **end if**
21: **end function**

---

samples can come from, our result has implications in the continuous setting as well. For instance, if we assume that there are (known) numbers $-\infty = y_0 < y_1 < \cdots < y_B = \infty$ such that in any interval $[y_i, y_{i+1}]$, the emperical CDF increases by at most $\alpha/2$, then we can again obtain quantile error $\alpha$ with $O(\frac{\log B}{\varepsilon^2 \alpha^2})$ users using our algorithm and bucketing users in the same interval $[y_i, y_{i+1}]$. As the dependency on $B$ in the number of samples is logarithmic, this might allow $B$ to be quite large, with a correspondingly small quantile error $\alpha$. We note that if the assumption on the CDF is incorrect, only the accuracy is affected while the algorithm remains private.

# I. Experiments

All experiments were carried out using an Intel Xeon Processor W-2245 (8 cores, 3.9GHz), 128GB RAM, Ubuntu 20.04.3, and Python 3.11. We considered the *success rate* for an error $\alpha$

$$\text{success rate} := \Pr\left[F_X(\tilde{m}) < \frac{1}{2} + \alpha \land F_X(\tilde{m} + 1) > \frac{1}{2} - \alpha\right],$$

where $F_X$ is the CDF of the sensitive dataset and $\tilde{m}$ is the median released by the algorithm, and the absolute quantile error

$$\text{error} := |F_X(\tilde{m}) - F_X(m)|,$$

as the main metrics for our evaluation. Each algorithm was executed 200 times to compute the empirical cumulative distribution of the absolute quantile error. As error we opted for the standard deviation of the sample average success rate, calculated as:

$$\sigma = \sqrt{\frac{\tilde{p}(1-\tilde{p})}{200}} \qquad \text{where} \quad \tilde{p} = \frac{1}{200}\sum_{i=1}^{200}\left[F_X(\tilde{m}_i) < \frac{1}{2} + \alpha \land F_X(\tilde{m}_i + 1) > \frac{1}{2} - \alpha\right]$$

**Data Generation**  The income dataset was generated using a Pareto distribution $p(x) \sim \frac{1}{x^{\gamma+1}}$, a well studied distribution to model income data (Arnold, 2014). We generate $n = \{2500, 5000, 7500\}$ positive integers by sampling from the continuous Pareto distribution with shape $\gamma = 1.5$ and multiplicative factor 2000 and then rounding them. For different coin domains $[B]$ we clip the dataset to get integer values in $[B]$. To compare DpBayeSS and DpNaiveNBS across various coin domains $[B]$ for a fixed privacy budget, we generated $n = 2500$ integers by sampling from a uniform distribution over a random interval within $[B]$. This approach avoids having the median around $B/2$, which would make the problem too straightforward.

**Implementation Details**  These mechanisms are run across the entire data set, which means that each user is sampled once.

- DpNaiveNBS is a standard differentially private implementation of noisy binary search introduced in (Karp & Kleinberg, 2007), where each coin flip is privatized using randomized response with $\varepsilon$ privacy budget, which we call $RR_\varepsilon$. It searches for the coin with probability closest to $\frac{1}{2}$ using a standard binary search. To estimate a coin probability, it samples without replacement batches of size $b = \lfloor \frac{n}{\lceil \log_2 B \rceil} \rfloor$, then redistributes the remaining samples $n - \lceil \log_2 B \rceil b$ by adding one sample to each batch starting from the first. Due to randomized response, any empirical probability $p_c = \frac{1}{b} \sum_{x \in \text{batch}} RR_\varepsilon([x \leq c])$ is unbiased $\tilde{p}_c = \frac{e^\varepsilon+1}{e^\varepsilon-1}(p_c - \frac{1}{e^\varepsilon+1})$ before confronting it with $\frac{1}{2}$.

- DpBayeSS is a implementation of Algorithm 3 in (Gretta & Price, 2024), with some minor changes ($\gamma$ is set to $1/13$ in line 11 of Algorithm 3), where each coin flip is privatized using randomized response. The algorithm runs at most two DpBayesLearn (a differentially private implementation of Algorithm 2 where each coin flip is privatized using randomized response) and further makes use of DpNaiveNBS on the remaining coins to get the one with head probability closest to $\frac{1}{2}$. The sample budget $n$ is divided into $M_{B_1}$, $M_{B_2}$ for the two DpBayesLearn and $M_S$ for the DpNaiveNBS. The split satisfies the following ratios suggested in (Gretta & Price, 2024) $\frac{M_{B_1}}{M_{B_2}} = \frac{\log B}{\log \log B}$, $\frac{M_{B_1}}{M_S} = \log B$, and $\frac{M_{B_2}}{M_S} = \log \log B$.

  BayesLearn is designed to take $\alpha$ as an input for updating the weights during the Bayesian learning step (see Algorithm 2), hence it assumes a sufficiently large number of users. To reverse this approach, so using $n$ as input, we empirically determine the minimum value of $\alpha$ achievable by the algorithm. For DpBayesLearn and $\varepsilon < 1$ we showed that to get an error $\alpha$ we need to solve for an error $\tilde{\alpha} = \frac{\alpha \varepsilon}{8}$. For a fixed $n$, the error cannot be smaller than $\alpha \geq 8c \frac{\sqrt{\log B}}{\varepsilon \sqrt{n}}$ for some constant $c > 0$, therefore we have that $\tilde{\alpha} \geq c\sqrt{\frac{\log B}{n}}$. We analyze different values of $c$ in the hyper-parameter selection section in order to get a value of $c$ such that the algorithm, run with $\tilde{\alpha} = c\sqrt{\frac{\log B}{n}}$, gives better results. For completeness and full reproducibility we provide a pseudocode of our implementation in Algorithm 4

- Hierarchical Mechanism is built according to (Cormode et al., 2019). Essentially, we constructed a tree with branching factor equal to 4 and at each level we store the 4-adic decomposition of $[B]$. For example, if $B = 4^9$ at the first level of the tree, composed by four nodes, is stored $\{[1, \ldots, 4^8], [4^8 + 1, \ldots, 2 \cdot 4^8], [2 \cdot 4^8 + 1, \ldots, 3 \cdot 4^8], [3 \cdot 4^8 + 1, \ldots, 4^9]\}$, while on the leaves are stored all the possible singletons. Each user selects a random level of the tree and reports its position using *unary encoding* (Wang et al., 2017; Cormode et al., 2021). For this LDP protocol we used the public library at the following GitHub repository[8]. After filling the tree, we computed the whole cumulative distribution and released the coin with closest value to $\frac{1}{2}$.

---

[8] https://github.com/Samuel-Maddock/pure-LDP

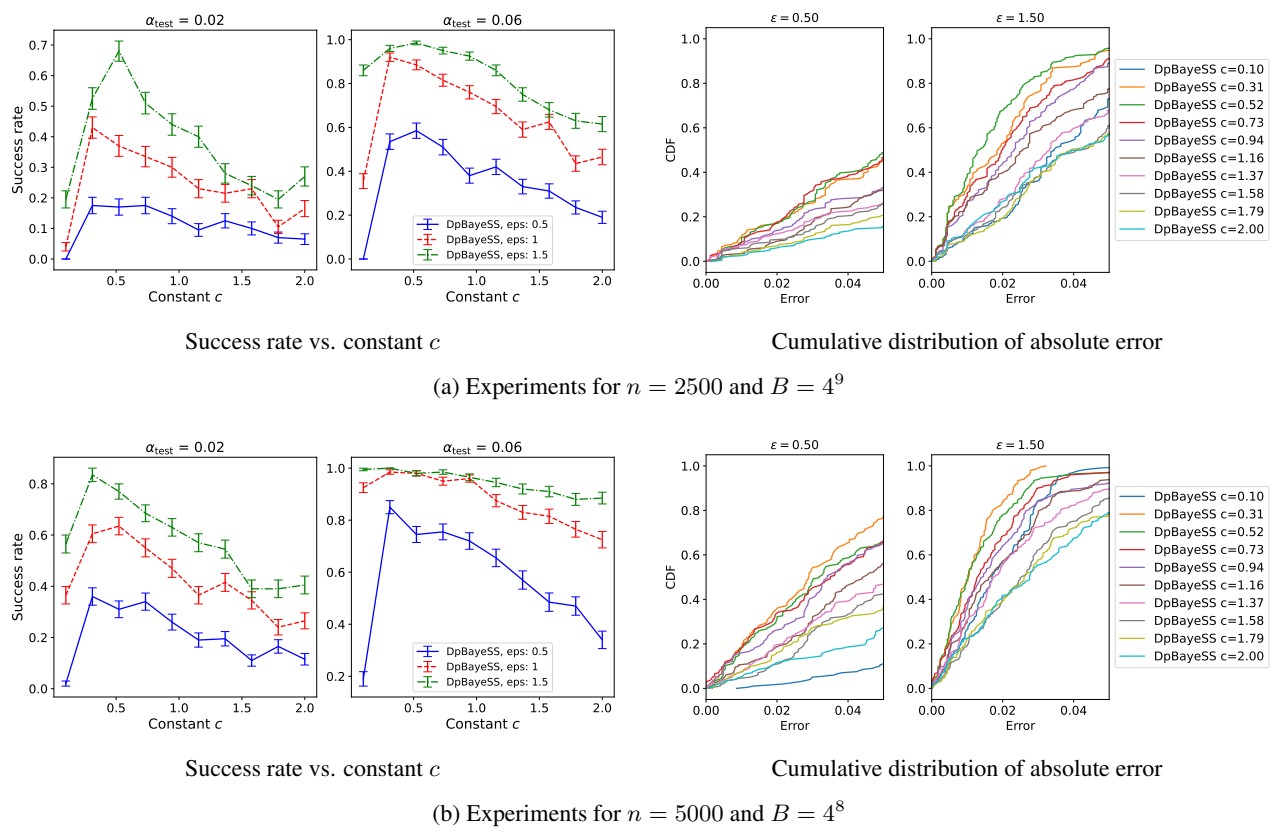

(a) Experiments for $n = 2500$ and $B = 4^9$

(b) Experiments for $n = 5000$ and $B = 4^8$

*Figure 2.* Experiments to estimate the best constant $c$ to compute $\alpha_{\text{update}} = c\sqrt{\frac{\log B}{n}}$.

## I.1. Hyper-Parameter Selection

To determine the optimal parameter for updating `DpBayesLearn` given a fixed number of users $n$, coins $B$, and varying privacy budgets $\varepsilon \in \{0.5, 1, 1.5\}$, we conducted experiments using `DpBayeSS` with different update parameters $\alpha_{\text{update}} = c\sqrt{\frac{\log B}{n}}$. These experiments were performed on two distinct datasets generated by sampling from a Pareto distribution, the outcomes are illustrated in Figure 2. By analyzing different $\alpha_{\text{test}}$ values and the error distribution across various privacy budgets, we observed that the algorithm performs poorly at $c = 0.1$. However, within the range $c \in [0.3, 0.7]$, the performance stabilizes and the precision decreases for $c > 0.7$. Therefore, an effective range for the parameter $c$ is $[0.3, 0.7]$. Based on this analysis, we chose to use $c = 0.6$. We note that our analysis is tailored to the high-privacy regime; however, in practice, this constant also yields a well-performing algorithm for $\varepsilon < 5$.

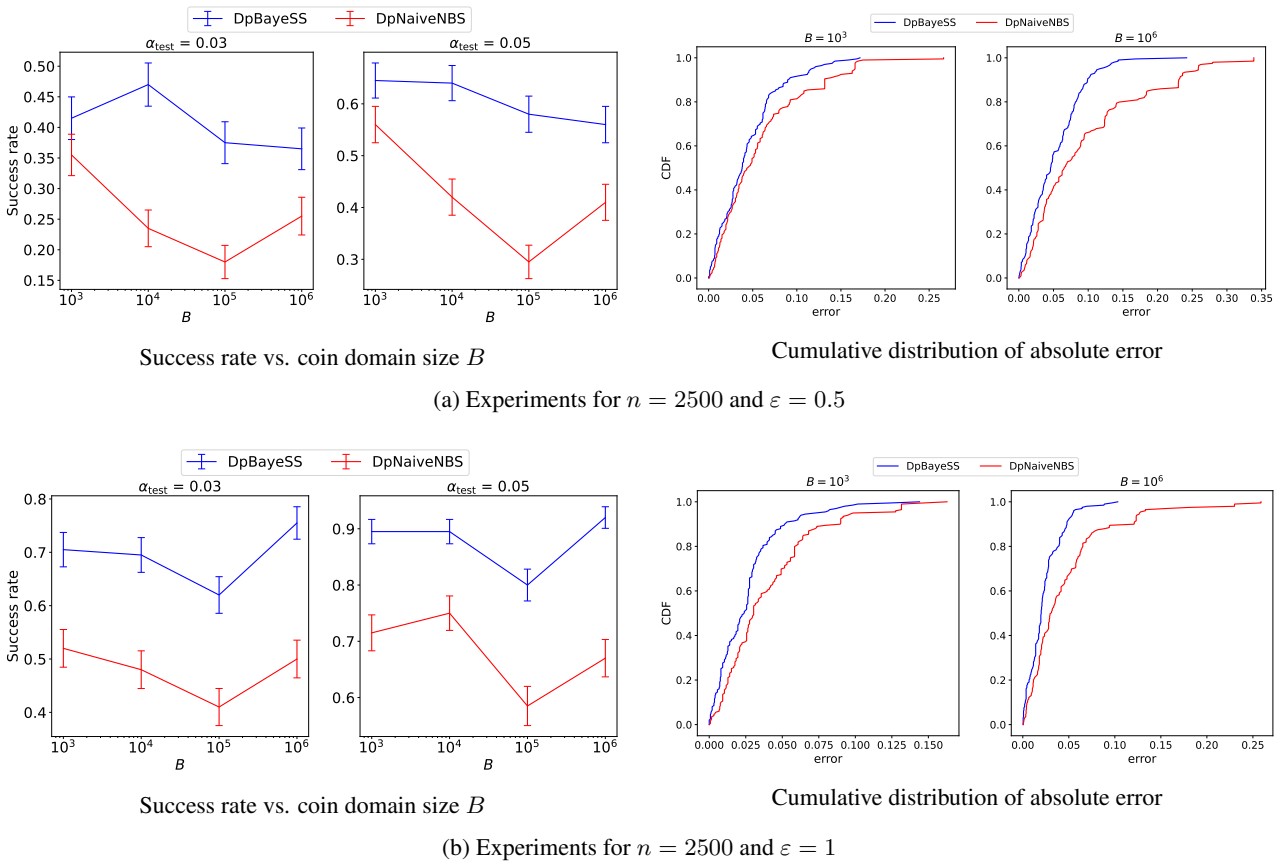

(a) Experiments for $n = 2500$ and $\varepsilon = 0.5$

(b) Experiments for $n = 2500$ and $\varepsilon = 1$

*Figure 3.* Experiments run over a dataset obtained by sampling $n$ random integers over a random subset of $[B]$.

## I.2. Comparison Analysis

In Figure 5 we run the three algorithms two Pareto like dataset with $n = \{2500, 7500\}$ and $B = \{4^9, 4^8\}$ with various privacy budgets $\varepsilon \in [0.1, 5]$. We observed how the adaptive mechanisms `DpBayeSS` and `DpNaiveNBS` outperform a non adaptive mechanism such as `Hierarchical Mechanism`. This superior performance is not surprising as the former algorithms are tailored specifically for median estimation. In contrast, `Hierarchical Mechanism` constructs a differential private data structure capable of answering any range queries with an error of polylog($B$). From our results it is clear the `DpBayeSS` is more likely to return a coin with low quantile error than `DpNaiveNBS`, both for $\varepsilon < 1$ and $\varepsilon > 1$. This result aligns with the findings in (Gretta & Price, 2024), where the authors conducted experiments demonstrating that `BayeSS` can achieve the same error rate as `NaiveNBS` (algorithms without randomized response) using fewer coin flips and, consequently, fewer user samples.

We conducted further experiments to evaluate the behavior of `DpBayeSS` and `DpNaiveNBS` for different coin domains $[B]$. The dataset is obtained by sampling $n = 2500$ integers uniformly from a random interval in $[B]$, for any $B \in \{10^3, 10^4, 10^5, 10^6\}$. The main results are listed in Figure 3 for two different privacy budgets $\varepsilon \in \{0.5, 1\}$. We observed that `DpBayeSS` is more stable than `DpNaiveNBS` for different coin domains, and offers good utility for realistic privacy budget and error (e.g. for $\alpha_{\text{test}} = 0.05$, $\varepsilon = 1$, and $n = 2500$, `DpBayeSS` returns a $\alpha_{\text{test}}$-good median with probability higher than $0.8$).

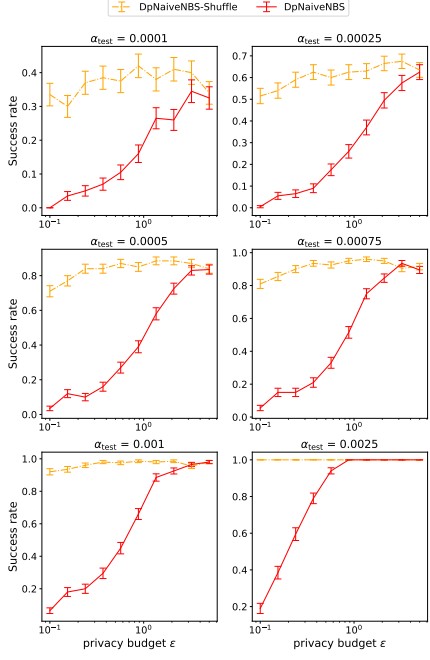
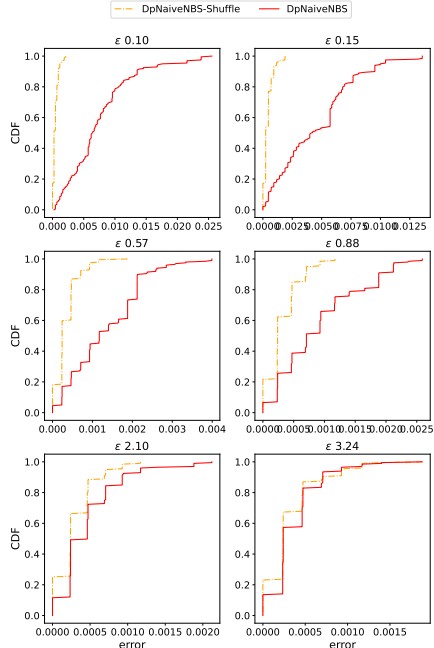

Success rate vs. privacy budget $\varepsilon$ | Cumulative distribution of absolute error

*Figure 4.* Experiments for $n = 10^7$ and $B = 4^8$, with $\delta = 10^{-8}$ for shuffle DP

### I.3. Noisy Binary Search with Shuffling

When the number of users $n$ is sufficiently large, noisy binary search with shuffling, as described in Section G, can be implemented. The implementation mirrors that of `DpNaiveNBS`, but the privacy budget $\varepsilon_{\mathrm{RR}}$ for randomizing the coin flip is determined as $\varepsilon_{\mathrm{RR}} = \log\left(\frac{\varepsilon^2}{80\log(4/\delta)}\left(\left\lfloor\frac{n}{\lceil\log_2 B\rceil}\right\rfloor + 1\right)\right)$ to achieve $(\varepsilon, \delta)$-differential privacy (DP) under shuffling, as established in Lemma G.5. Since $\varepsilon_{\mathrm{RR}} \geq 0$ is required, the user population $n$ must be sufficiently large to enable this amplification technique. In particular, for $\delta = 10^{-8}$, $\varepsilon \in \{0.1, 0.5, 1\}$, and $B = 4^8$, it is necessary to have $n$ greater than $2.5 \cdot 10^7, 7.8 \cdot 10^4$ and $1.3 \cdot 10^4$, respectively, making a direct comparison with the previous experiments impossible. However, we note that the required number of users can be significantly reduced in practice by employing a tighter numerical upper bound for privacy amplification by shuffling (Feldman et al., 2021).

We generated a Pareto-like dataset (as described in the Data Generation section) with $n = 10^7$ and conducted experiments using $\delta = 10^{-8}$ and various privacy budgets $\varepsilon \in [0.1, 5]$. Due to limited computational resources and the non optimal implementation of `DpBayesSS` and `Hierarchical Mechanism`, we restricted our comparison to `DpNaiveNBS` and its variant implemented with privacy amplification through shuffling. The results are shown in Figure 4. For small privacy budgets, shuffling-based amplification provides higher utility, whereas for $\varepsilon > 3$, the performance of the algorithms converges and becomes comparable.

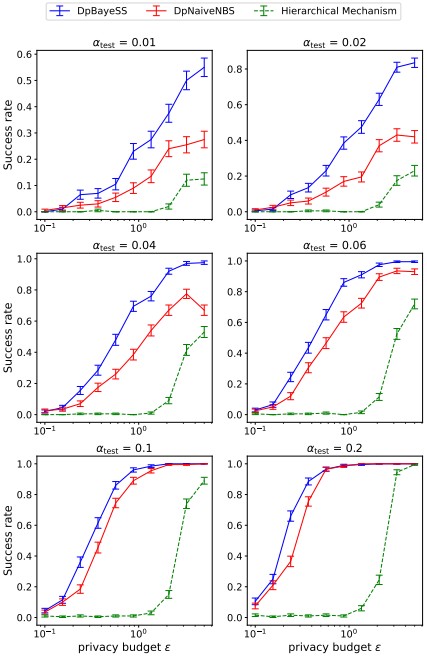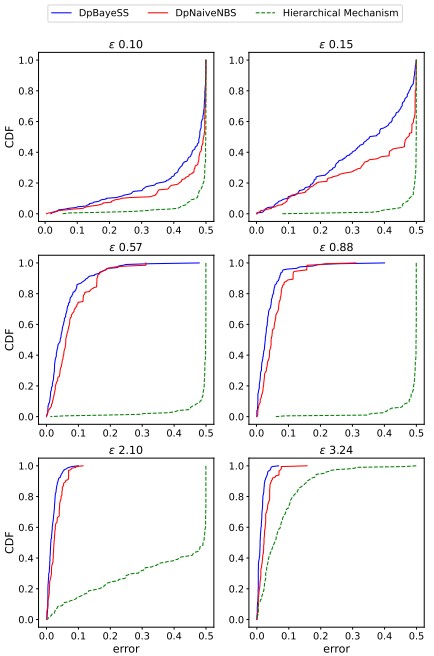

Success rate vs. privacy budget $\varepsilon$          Cumulative distribution of absolute error

(a) Experiments for $n = 2500$ and $B = 4^9$

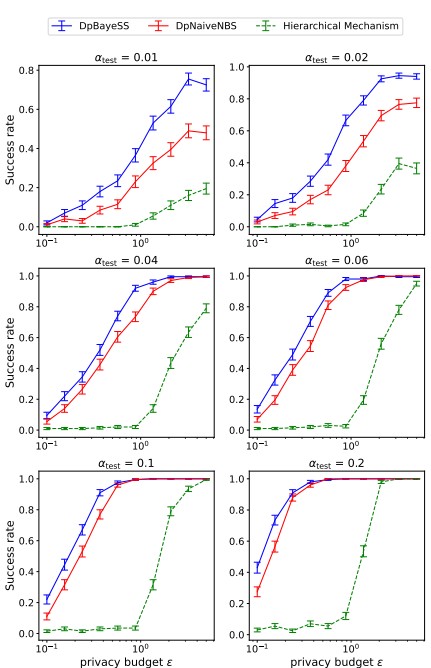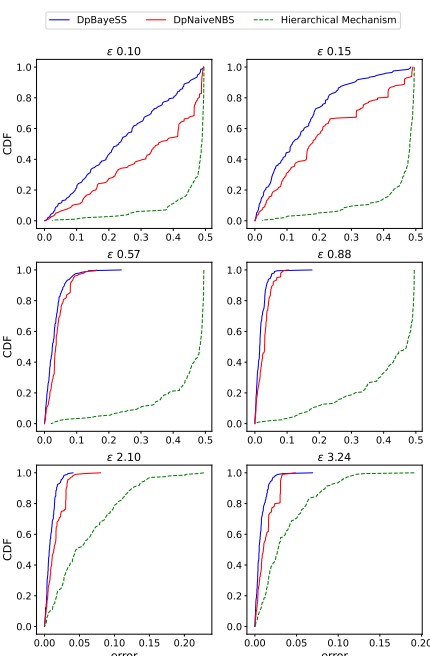

Success rate vs. privacy budget $\varepsilon$          Cumulative distribution of absolute error

(b) Experiments for $n = 7500$ and $B = 4^8$

*Figure 5.* Comparison analysis.

