# OpenReview forum: "Lightweight Protocols for Distributed Private Quantile Estimation"
_ICML.cc/2025/Conference — ICML 2025 spotlightposter_

### Official Review · Reviewer_KDYn · 2025-03-05

**Overall Recommendation:** 4

**Summary:**

The authors study the problem of estimating quantiles under local differential privacy and under shuffle differential privacy, with applications in distributed and private quantiles estimation.
To do so, the paper presents new algorithms.
The article presents both upper and lower bounds for the problems at hand.
Finally, the article compares experimentally the new methods that it introduces with already existing methods.

## update after rebuttal

After the rebuttal, I will maintain my original score.

**Claims And Evidence:**

The claims of the article are well-supported by clear proofs and empirical evaluation.

**Essential References Not Discussed:**

NA

**Experimental Designs Or Analyses:**

I checked the soundness of the experiments that are presented in the main body of the article. I did not however check the correctness of the code that was provided by the authors.

**Methods And Evaluation Criteria:**

The methods and evaluation criteria make sense for the problem at hand.

**Other Comments Or Suggestions:**

NA

**Other Strengths And Weaknesses:**

Strengths :
- The article is theoretically sound and overall well written.
- The problem that is tackled is of interest for the community of privacy preserving machine learning.

I overall have a positive opinion of the paper, so the following weaknesses are more in order to improve the paper than they are to point out reasons to reject it.

Weaknesses :

- The structure of the paper makes it slightly difficult to follow, as the technical details of the algorithm are introduced quite late. This forces the reader to jump back and forth to understand how the challenge outlined in the introduction is actually addressed. Adding brief “how-to” explanations earlier could improve readability.
- The definition of an approximate median in Definition 2.2 could be clarified. While it is defined for empirical distributions in the introduction, it is not explicitly extended to general distributions. A quick reminder of this concept in Definition 2.2 would be helpful.
- The article would benefit from a brief conclusion summarizing key insights, highlighting open questions, and outlining possible directions for future work.

**Questions For Authors:**

1. The study focuses on "quantile error," which measures how much probability mass the estimate is off by. However, in some applications, the primary concern is the uncertainty on the actual value (especially when the grid is a discretization of the real line and inherits a distance). Can the results be extended to this setting?
2. In Theorem 1.4, the condition n=O(…) suggests that the procedure works as long as there isn’t too much data. This seems counterintuitive, as one might expect more data to make the problem easier. Could you clarify this?
3. In the same theorem, why do 𝛼 and 𝜖 appear to interact independently in the bound? Is this a consequence of the specific regime being considered?
4. The paper states: “We are typically interested in the high-probability setting, where β=1/poly(B).” Is this because setting β=1/B would be equivalent to making a random guess? If so, adding a sentence to explain this, and clarifying that the polynomial must be greater than B for relevance, could be helpful.

**Relation To Broader Scientific Literature:**

The authors link their work to prior literature and claim (with theoretical evidences) that their new algorithm improves prior state of the art by a polylogarithmic factor in the size of the set on which they work.

**Theoretical Claims:**

I checked the proofs that are presented in the main body of the article, and I looked at the proof techniques that are used in the appendix. No issue was found, yet I did not check every single detail.

---

> ### Author Rebuttal · Authors · 2025-03-31
>
> Thanks for your questions and valuable feedback!
>
> > The structure of the paper makes it slightly difficult to follow, as the technical details of the algorithm are introduced quite late. This forces the reader to jump back and forth to understand how the challenge outlined in the introduction is actually addressed. Adding brief “how-to” explanations earlier could improve readability.
>
> We will do our best to add more such explanations. Feel free to let us know if there was any part that was particularly confusing.
>
>
> >The definition of an approximate median in Definition 2.2 could be clarified. While it is defined for empirical distributions in the introduction, it is not explicitly extended to general distributions. A quick reminder of this concept in Definition 2.2 would be helpful.
>
> Thanks for pointing this out. We will clarify in the paper.
>
> >The article would benefit from a brief conclusion summarizing key insights, highlighting open questions, and outlining possible directions for future work.
>
> Thanks for the suggestion. We agree, and will include it in the next version of the paper.
>
> >The study focuses on "quantile error," which measures how much probability mass the estimate is off by. However, in some applications, the primary concern is the uncertainty on the actual value (especially when the grid is a discretization of the real line and inherits a distance). Can the results be extended to this setting?
>
> Our results cannot be generalized to a domain-based error without making an assumption on the underlying data distribution. Lower bounds from [1] show that any median estimation protocol must have error that grows linearly with B when error is measured with the average distance from the estimate to each data point. Additionally note that minimizing the distance in the domain between the estimated and true median must also have error $\Omega(B)$, even under central DP. Indeed, if $n$ is odd and $(n+1)/2$ of the $x_i$’s are 0 and the remaining $(n-1)/2$ $x_i$’s are $B$, then the median is $0$, but changing the data of just a single user can change the median to $B$. We expect that similar lower bounds would hold for other data-domain based error functions.
>
>
> [1] Duchi, J. C., Jordan, M. I., & Wainwright, M. J. (2018). Minimax Optimal Procedures for Locally Private Estimation. Journal of the American Statistical Association, 113(521), 182–201. https://doi.org/10.1080/01621459.2017.1389735
>
> >In Theorem 1.4, the condition $n=O(\dots)$ suggests that the procedure works as long as there isn’t too much data. This seems counterintuitive, as one might expect more data to make the problem easier. Could you clarify this?
>
> Thanks for pointing this out. The statement should be that only $n=O(\dots)$ users are needed to get an $\alpha$-approximate median. We will update the writing.
>
> >In the same theorem, why do $\alpha$ and $\epsilon$ appear to interact independently in the bound? Is this a consequence of the specific regime being considered?
>
> In the following, we ignore logarithmic factors and focus on the polynomial dependence on $\alpha$ and $\epsilon$. In order to apply privacy amplification by shuffling (Lemma G.5), we have to pick the number of users in a batch $n’\gg 1/\epsilon^2$. With this many users, shuffle DP ensures that essentially each user in the batch answers threshold queries correctly with probability $>3/4$. With $>3/4$ probability of correct answers, it suffices to have $O(1/\alpha^2)$ users in a batch to estimate the fraction of these users with $x_i$’s below a given threshold $t$ within $O(\alpha)$. Additionally, there is another reason that the batch has to be this big, namely to ensure that the threshold query of the batch is within $O(\alpha)$ of the threshold query of the full data set. Together, this gives sample complexity $O(1/\alpha^2+1/\epsilon^2)$. The additional logarithmic dependencies come from the fact that these events have to hold with less than constant error probability (since we union bound over $\lg B$ steps of the binary search). Finally, we need $\lg B$ times as many users, since there are $\lg B$ steps to the binary search.
>
> >The paper states: “We are typically interested in the high-probability setting, where β=1/poly(B).” Is this because setting β=1/B would be equivalent to making a random guess? If so, adding a sentence to explain this, and clarifying that the polynomial must be greater than B for relevance, could be helpful.
>
> We note that $\beta$ is the \emph{failure} probability. A random guess would thus give $\beta=1-\Theta(\alpha)$, since the probability of guessing an $\alpha$-approximate median is $\Theta(\alpha)$. Our algorithm guarantees that our median estimate is bad only with very low probability $1/poly(B)$, say $1/B^{10}$.

---

> > ### Comment · Reviewer_KDYn · 2025-04-04
> >
> > Thanks to the authors for their response and for integrating the modifications that they mentioned in the paper. I will maintain my score.

---

### Official Review · Reviewer_EwQE · 2025-03-06

**Overall Recommendation:** 3

**Summary:**

This paper considers the estimation of quantiles under the LDP framework with bounded integral data. It derives a series of lower bounds under both shuffle-DP and LDP, and proposes an LDP algorithm in an adaptive setting.

**Claims And Evidence:**

Yes.

**Essential References Not Discussed:**

No.

**Experimental Designs Or Analyses:**

Yes.

**Methods And Evaluation Criteria:**

Yes.

**Other Comments Or Suggestions:**

See the questions below.

**Other Strengths And Weaknesses:**

Pros: Quantile estimation under LDP is a significant problem and the non-asymptotic results appear to be interesting.

Cons: The authors assume data are drawn from a bounded integral space (with finitely many values), which seems fairly restrictive and uncommon.

**Questions For Authors:**

1. For random variables with infinitely many possible values (e.g., Poisson), does the proposed algorithm or framework still apply?

2. Regarding footnote 1, I may be mistaken, but I believe some prior work considers privacy-preserving quantile estimation on a continuous interval (e.g., on $[0,1]$), such as [1] in CDP setting and [2] in the LDP setting. Could the authors clarify how these results relate to their work?


3. The paper specifies a particular range for $\epsilon$. For very large privacy budgets (i.e., $\epsilon \to \infty$), will the results degrade or converge to the non-private results (as seen in [3] and [4])? In other words, do these bounds become consistent with classical results in the absence of privacy constraints?


[1] Lalanne, C., Garivier, A., & Gribonval, R. (2023). Private statistical estimation of many quantiles. In International Conference on Machine Learning (pp. 18399-18418). PMLR.

[2] Liu, Y., Hu, Q., Ding, L., & Kong, L. (2023). Online local differential private quantile inference via self-normalization. In International Conference on Machine Learning (pp. 21698-21714). PMLR.

[3] Chen, L., Keilbar, G., & Wu, W. B. (2023). Recursive quantile estimation: Non-asymptotic confidence bounds. Journal of Machine Learning Research, 24(91), 1-25.

[4] Howard, S. R., & Ramdas, A. (2022). Sequential estimation of quantiles with applications to A/B testing and best-arm identification. Bernoulli, 28(3), 1704-1728.

**Relation To Broader Scientific Literature:**

See the pros below.

**Theoretical Claims:**

Yes.

---

> ### Author Rebuttal · Authors · 2025-03-31
>
> Thanks for your questions and your consideration of our paper
>
> >For random variables with infinitely many possible values (e.g., Poisson), does the proposed algorithm or framework still apply?
>
> Without any assumptions on the distribution of the random variable, known lower bounds (as we discuss under related work) show that it is impossible to obtain any meaningful error guarantees, even in the central setting of DP. For concrete distributions (like Poisson with a reasonable bound on its rate, or even continuous distributions), one can usually apply our algorithm combined with bucketing as we discuss in Appendix H. The error guarantees will then depend on how much the CDF can change within a single bucket which in turn depends on the parameter space of the class of distributions. While it is interesting how well classes of infinite or continuous distributions can be discretized, this direction of work is somewhat orthogonal to ours.
>
> >Regarding footnote 1, I may be mistaken, but I believe some prior work considers privacy-preserving quantile estimation on a continuous interval (e.g., on ), such as [1] in CDP setting and [2] in the LDP setting. Could the authors clarify how these results relate to their work?
>
> [1] considers continuous distributions with the same error function as ours. However, related to our discussion in Appendix H, their results depend on a parameter $\Delta$ that expresses how well the continuous distribution can be discretized into a finite set of intervals on which the CDF does not increase too quickly. [2] extends a line of work (see e.g., Duchi et al. [3]) aiming to minimize the average distance between the estimated median and the data points (and a more generalized form for other quantiles). Lower bounds in [3] show that this error must grow linearly with the range in which the true median lies. In our setting, without further assumptions on the data distribution, this would be linear in $B$. We additionally note that minimizing the distance in the domain between the estimated and true median must also have error $\Omega(B)$, even under central DP. Indeed, if $n$ is odd and $(n+1)/2$ of the $x_i$’s are 0 and the remaining $(n-1)/2$ $x_i$’s are $B$, then the median is $0$, but changing the data of just a single user can change the median to $B$.
>
> On the other hand, our work aims to minimise the *rank error* of the quantile and is not subject to these lower bound.
>
>
> [1] Lalanne, C., Garivier, A., & Gribonval, R. (2023). Private statistical estimation of many quantiles. In International Conference on Machine Learning (pp. 18399-18418). PMLR.
>
> [2] Liu, Y., Hu, Q., Ding, L., & Kong, L. (2023). Online local differential private quantile inference via self-normalization. In International Conference on Machine Learning (pp. 21698-21714). PMLR.
>
> [3] Duchi, J. C., Jordan, M. I., and Wainwright, M. J. Minimax optimal procedures for locally private estimation. Journal of the American Statistical Association, 113(521):182– 201, 2018.
>
> >The paper specifies a particular range for $\epsilon$. For very large privacy budgets (i.e., $\epsilon\to \infty$), will the results degrade or converge to the non-private results (as seen in [3] and [4])? In other words, do these bounds become consistent with classical results in the absence of privacy constraints?
>
> Our result states that $O((\log B)/\alpha^2)$ users suffice to find an $\alpha$-approximate median under LDP with high probability in $B$ even for $\epsilon=O(1)$.iThis matches known lower bounds even absent privacy constraints for statistical median estimation (when  requiring high probability in $B$). On the other hand if we just want success probability $2/3$, absent privacy constraints, $O(1/\alpha^2)$ users suffice for statistical median estimation. It is an interesting open question to design a protocol with the correct convergence to $O(1/\alpha^2)$ as $\epsilon\to \infty$, but would likely require quite different techniques than the ones presented in our paper.
>
> In the *empirical* setting, absent privacy constraints, an algorithm can simply output the median of the data set, so one should expect that as $\epsilon\to \infty$ the required number of users would converge to $1$. Again, it is interesting to understand this convergence. In any case, we focus on the common choice of $\epsilon=O(1)$.

---

> > ### Comment · Reviewer_EwQE · 2025-04-07
> >
> > I appreciate the authors' response and their efforts to address my concerns. It is indeed different to derive non-asymptotic rank error results, as compared to mean absolute error, which is a common choice in applications (as another reviewer also mentioned). This paper indeed presents many nontrivial results about LDP quantile estimation, except for the finite assumption about $B$, because the entire framework is based on given data sets rather than on potential distributions. Therefore, I will raise the score to 3. However, I still suggest that the authors add more explanation in final version, regarding the proposed setting and error metrics, especially the rationale for choosing this particular quantile error and the additional challenges it poses compared to the traditional setting.

---

### Official Review · Reviewer_HTT3 · 2025-03-13

**Overall Recommendation:** 4

**Summary:**

This paper studies quantile estimation under local differential privacy. They are interested in the sequentially adaptive local model, where the aggregator queries each user only once, but in rounds where the set of users and the randomizer they are asked to use can depend on information learned in previous rounds.

They first argue via folklore tricks that estimating any quantile can be reduced to estimating the median with twice as many users. Hence, the main task is to estimate the median of either a distribution (where each user samples one example in an i.i.d. fashion from it), or of a dataset from a discrete domain B. They give a sequentially adaptive $\epsilon$-DP $n$-round algorithm in the local model that outputs the $\alpha$-approx median whp, as long as the number of users is $O(\log B / \alpha^2 \epsilon^2)$. This is the main contribution of the paper and follows by using the noisy binary search primitive considered in prior work (Gretta and Price 2024) where you have a number of coins with monotonically increasing success probabilities, a target probability, and can flip coins, with the goal of finding a pair of successive coins containing the target probability. The statistical setting is a straightforward reduction to this problem, but the empirical setting is more challenging since you can't simply sample with replacement from the empirical distribution (since then you might see the same user multiple times- resulting in an algorithm that is not sequentially adaptive). The authors instead use sampling without replacement, along with randomly permuting the users, and argue via martingale-based techniques that for any threshold in $[B]$, and any $t > 0$, the probability that we draw one of the remaining $n-t$ users that are less than the threshold remains almost unchanged from what it was when $t=0$. They then argue that the techniques of Gretta and Price for the noisy binary search problem satisfy a robustness property;  they work even if we don't flip exactly the same coins, but rather ones with close probabilities. This is sufficient to solve the problem in the empirical setting.

They also show a matching lower bound for sequentially adaptive local model algorithms using Fano's inequality-based techniques of Duchi et al. They also appeal to a result of Edmonds, Nikolov, and Ullman to argue that this gives a separation from the non-adaptive local model, where there is a lower bound of $\Omega(log^2 B / \epsilon^2)$ for this problem (for constant $\alpha$). They also give a result in the shuffle model that achieves better round complexity than the $n$-round local model protocol (with similar total number of users).

**Claims And Evidence:**

The claims seem correct and are largely well explained.

**Essential References Not Discussed:**

Not essential, but Cohen et al. (STOC 2023) is the latest paper on quantile estimation/interior point in the central model that should be cited in the related work (where Kaplan et al. and others are mentioned) since it gets asymptotically tight bounds. Also I'd add in a more detailed exposition of work on adaptivity (and separations with non-adaptivity) in the local model (for e.g. Joseph Mao Neel Roth 2019, Joseph Mao Roth 2020, Daniely Feldman 2019, Acharya Canonne Sun Tyagi 22 for e.g.).

**Experimental Designs Or Analyses:**

Yes, see methods and evaluation criteria.

**Methods And Evaluation Criteria:**

The primary results are theoretical and experiments are secondary considerations in the paper - however, they do consider experiments where they compare their method to two other more standard baseline algorithms, and run on datasets of different sizes, drawn from two types of distributions- pareto and uniform. They evaluate the absolute quantile error and success rate (the fraction of times a good quantile is released). They also choose reasonable $\epsilon$ values in their evaluations. Overall, the experimental setup makes sense.

**Other Comments Or Suggestions:**

1) instead of adversarial monotonic NBS I'd call the problem robust monotonic NBS (adversarial makes it sound like an adversary can choose coins maliciously).

2) The indices in Lemma 3.1 and 4.1 are overloaded (i and j) which was confusing on first read. Also in Lemma 4.1 the set in the theorem statement should have c_j = 0 not c_i=0.

3) in the proof of Lemma 3.1, the comment on conditioning on pi(1),...pi(i) is confusing because conditioning on X_js does not imply conditioning on pi(1)....pi(i) (but you don't need to condition on that part of the permutation for the argument to go through).

**Other Strengths And Weaknesses:**

This paper cleverly uses existing tools, and has to jump through a number of technical hoops in order to get them to work for empirical quantile estimation (in the sequentially adaptive local model). Demonstrating the robustness of the Gretta-Price noisy binary search algorithm may be of independent interest.

**Questions For Authors:**

See questions on shuffle model in the 'theoretical claims' section.

**Relation To Broader Scientific Literature:**

Prior work has obtained non-adaptive algorithms for quantile estimation in the local model of differential privacy with suboptimal dependence on the domain size; this is the first work to get the correct dependence (leveraging adaptivity). This work fits in the larger literature on the local model of differential privacy (using tools in the model like randomized response and mutual information-based lower bonds), as well as the larger literature on private quantile estimation (extensively explored for the central model of differential privacy as well and a fundamental problem both theoretically and practically).

**Theoretical Claims:**

Yes, I checked all the proofs for correctness, and am convinced that the most significant results are correct. The only one I was more unsure of was the shuffle model result, where I didn't fully understand the algorithm that they were using (the proof of Theorem 1.4 is rather vague on this point so fully specifying the algorithm would make it easier to verify).

---

> ### Author Rebuttal · Authors · 2025-03-31
>
> Thanks for your interest in our paper and your valuable feedback!
>
> > The only one I was more unsure of was the shuffle model result, where I didn't fully understand the algorithm that they were using (the proof of Theorem 1.4 is rather vague on this point so fully specifying the algorithm would make it easier to verify).
>
> Thanks for pointing this out. We agree that the writing surrounding the proof of theorem 1.4 can be improved. We will update the paper by describing the algorithm in detail and including pseudocode before proving the theorem in the appendix. To clarify, the algorithm randomly partitions the users into batches of size roughly $\frac{1}{\epsilon^2}+\frac{1}{\alpha^2}$. For a given threshold $t\in [B]$, our analysis then shows that the shuffled randomized responses to whether the users are above or below $t$ suffice to determine the cdf at $t$ within additive $\alpha$ with sufficiently high probability. Our algorithm thus uses $\log_2(B)$ such batches to perform a binary search on $t$. This is similar to the algorithm given by Karp and Kleinberg [1] (section 1.2), but with extra care needed to ensure privacy.
>
> > Not essential, but Cohen et al. (STOC 2023) is the latest paper on quantile estimation/interior point in the central model that should be cited in the related work (where Kaplan et al. and others are mentioned) since it gets asymptotically tight bounds. Also I'd add in a more detailed exposition of work on adaptivity (and separations with non-adaptivity) in the local model (for e.g. Joseph Mao Neel Roth 2019, Joseph Mao Roth 2020, Daniely Feldman 2019, Acharya Canonne Sun Tyagi 22 for e.g.).
>
> Thanks for drawing these to our attention. We will include a discussion of these in the paper. In particular, we agree that a more detailed exposition on adaptivity in LDP should be included.
>
> > instead of adversarial monotonic NBS I'd call the problem robust monotonic NBS (adversarial makes it sound like an adversary can choose coins maliciously).
>
> We note that the algorithm for NBS works even in the case where an adversary can change coin probabilities maliciously (by at most some amount), hence the name. There is indeed no malicious adversary when we apply the NBS algorithm in our main protocol, but on the other hand, the changing coin probabilities have a complicated distribution. The strong adversarial assumption allows us to handle this, and it is unclear how to analyse our LDP protocol without such an assumption.
>
> >The indices in Lemma 3.1 and 4.1 are overloaded (i and j) which was confusing on first read. Also in Lemma 4.1 the set in the theorem statement should have c_j = 0 not c_i=0.
>
> Thanks for making us aware of this, we will fix the $c_i$ typo and streamline the use of free variables in these lemmas.
>
> > in the proof of Lemma 3.1, the comment on conditioning on pi(1),...pi(i) is confusing because conditioning on X_js does not imply conditioning on pi(1)....pi(i) (but you don't need to condition on that part of the permutation for the argument to go through).
>
> Thanks for this point. In our first writeup, we defined the martingale as a Doob martingale, wrt the random choices of $\pi(1),...\pi(2n)$, but you are right that with the current definition, we should just condition on $(X_j)_{j<i}$. We will update the paper.

---

### Official Review · Reviewer_RF3G · 2025-03-14

**Overall Recommendation:** 4

**Summary:**

The paper studies the problem of finding quantiles with constraints of differential privacy. More specifically, it studies shuffle and local differential privacy.
The authors proved that the algorithms have utility higher than any known algorithm for the problem and also proved that the local DP algorithm’s bounds are tight and it is impossible to improve it further.

**Claims And Evidence:**

The evidence sufficiently supports the claims.

**Essential References Not Discussed:**

The paper doesn't lack any specific reference.

**Experimental Designs Or Analyses:**

The experimental design seems valid.

**Methods And Evaluation Criteria:**

Methods and evaluations are appropriate for the problem at hand.

**Other Comments Or Suggestions:**

N/A

**Other Strengths And Weaknesses:**

N/A

**Questions For Authors:**

N/A

**Relation To Broader Scientific Literature:**

The problem of estimating quantiles in differentially private way is well-studied in the literature since it is a common routine for may data analysis algorithms.
Local and shuffle differential privacy are also well-studied fields since they allow simplifying privacy guarantees of the systems using DP.
However, getting high utility from this algorithms is often hard.

Unfortunately, the LDP algorithms for quantile estimation were not studied enough, so I am really glad that this gap is getting closed.

**Theoretical Claims:**

The proofs presented in the paper are correct.

---

> ### Author Rebuttal · Authors · 2025-03-31
>
> Thanks for your interest in our paper!

---

### Decision · Program_Chairs · 2025-05-01

**Decision:**

Accept (spotlight poster)

**Comment:**

The reviews on this paper are uniformly positive, and all advocate acceptance.  The paper addresses an important problem in the context of machine learning, of privately gathering information on the distribution of data, in the form of quantiles.  They show improved results for this task over prior work in the distributed/federated setting, under local and shuffle differential privacy.  The reviewers appreciated the technical depth and importance of the work.  Some minor concerns were addressed clearly during the rebuttal phase, which led to the reviewers increasing their appreciation of the contribution.  On this basis, the paper is (strongly) recommended for acceptance in ICML.